# Evaluation of Blood Levels of C-Reactive Protein Marker in Obstructive Sleep Apnea: A Systematic Review, Meta‐Analysis and Meta-Regression

**DOI:** 10.3390/life11040362

**Published:** 2021-04-19

**Authors:** Mohammad Moslem Imani, Masoud Sadeghi, Farid Farokhzadeh, Habibolah Khazaie, Serge Brand, Kenneth M. Dürsteler, Annette Brühl, Dena Sadeghi-Bahmani

**Affiliations:** 1Department of Orthodontics, Kermanshah University of Medical Sciences, Kermanshah 6715847141, Iran; m.imani@kums.ac.ir; 2Medical Biology Research Center, Kermanshah University of Medical Sciences, Kermanshah 6715847141, Iran; msadeghi@kums.ac.ir; 3Students Research Committee, Kermanshah University of Medical Sciences, Kermanshah 6715847141, Iran; farid.farokhzadeh1398@yahoo.com; 4Sleep Disorders Research Center, Kermanshah University of Medical Sciences, Kermanshah 6715847141, Iran; hakhazaie@kums.ac.ir (H.K.); dena.sadeghibahmani@upk.ch (D.S.-B.); 5Center for Affective, Stress and Sleep Disorders (ZASS), Psychiatric University Hospital Basel, 4002 Basel, Switzerland; annette.bruehl@upk.ch; 6Department of Clinical Research, University of Basel, 4031 Basel, Switzerland; 7Department of Sport, Exercise and Health, Division of Sport Science and Psychosocial Health, University of Basel, 4052 Basel, Switzerland; 8Substance Abuse Prevention Research Center, Kermanshah University of Medical Sciences, Kermanshah 67146, Iran; 9School of Medicine, Tehran University of Medical Sciences, Tehran 25529, Iran; 10Psychiatric Clinics, Division of Substance Use Disorders, University of Basel, 4002 Basel, Switzerland; kenneth.duersteler@upk.ch; 11Center for Addictive Disorders, Department of Psychiatry, Psychotherapy and Psychosomatics, Psychiatric Hospital, University of Zurich, 8001 Zurich, Switzerland; 12Departments of Physical Therapy, University of Alabama at Birmingham, Birmingham, AL 35209, USA

**Keywords:** obstructive sleep apnea, inflammation, C-reactive protein, children, adults, meta-analysis

## Abstract

(1) Introduction: High sensitivity C-reactive protein (hs-CRP) and CRP are inflammatory biomarkers associated with several inflammatory diseases. In both pediatric and adult individuals with Obstructive Sleep Apnea (OSA) higher hs-CRP and CRP were observed, compared to controls. With the present systematic review, meta-analysis and meta-regression we expand upon previous meta-analyses in four ways: (1) We included 109 studies (96 in adults and 13 in children); (2) we reported subgroup and meta-regression analyses in adults with OSA compared to controls on the serum and plasma levels of hs-CRP; (3) we reported subgroup and meta-regression analyses in adults with OSA compared to controls on the serum and plasma levels of CRP; (4) we reported serum and plasma levels of both hs-CRP and CRP in children with OSA, always compared to controls. (2) Materials and Methods: The PubMed/Medline, Scopus, Cochrane Library, and Web of Science databases were searched to retrieve articles published until 31 May 2020, with no restrictions. The data included basic information involving the first author, publication year, country of study, ethnicity of participants in each study, age, BMI, and AHI of both groups, and mean and standard deviation (SD) of plasma and serum levels of CRP and hs-CRP. (3) Results: A total of 1046 records were retrieved from the databases, and 109 studies were selected for the analysis (96 studies reporting the blood levels of hs-CRP/CRP in adults and 13 studies in children). For adults, 11 studies reported plasma hs-CRP, 44 serum hs-CRP, 9 plasma CRP, and 32 serum CRP levels. For children, 6 studies reported plasma hs-CRP, 4 serum hs-CRP, 1 plasma CRP, and 2 serum CRP levels. Compared to controls, the pooled MD of plasma hs-CRP levels in adults with OSA was 0.11 mg/dL (*p* < 0.00001). Compared to controls, the pooled MD of serum hs-CRP levels in adults with OSA was 0.09 mg/dL (*p* < 0.00001). Compared to controls, the pooled MD of plasma CRP levels in adults with OSA was 0.06 mg/dL (*p* = 0.72). Compared to controls, the pooled MD of serum CRP levels in adults with OSA was 0.36 mg/dL (*p* < 0.00001). Compared to controls, the pooled MD of plasma hs-CRP, serum hs-CRP, plasma hs-CRP, and serum hs-CRP in children with OSA was 1.17 mg/dL (*p* = 0.005), 0.18 mg/dL (*p* = 0.05), 0.08 mg/dL (*p* = 0.10), and 0.04 mg/dL (*p* = 0.33), respectively. The meta-regression showed that with a greater apnea-hypapnea index (AHI), serum hs-CRP levels were significantly higher. (4) Conclusions: The results of the present systematic review, meta-analysis and meta-regression showed that compared to healthy controls plasma and serum levels of hs-CRP and serum CRP level were higher in adults with OSA; for children, and compared to controls, just plasma hs-CRP levels in children with OSA were higher.

## 1. Introduction

Repeated episodes of partial or complete obstruction of the airways during sleep characterize Obstructive Sleep Apnea (OSA) [1]. Polysomnographically measured Apnea–Hypopnea Index (AHI) as ≥5 [2,3] and AHI ≥ 1 events per hour [4] define OSA. A systematic review in 2017 found that the overall prevalence of OSA ranged from 9 to 38% of the general adult population, from 13 to 33% and 6 to 19% in males and females, respectively, with higher prevalence rates in elderly individuals [5]. Therefore, OSA is more prevalent in males than females [6], and OSA increases with age in adults [7]. Further, OSA is related to overweight: the prevalence of OSA varies from 60% to 70% in obese people, and reaches more than 90% in individuals with severe obesity [8,9]. OSA may be associated with structural brain alternations [10], and two recent meta-analyses showed that compared to healthy controls inflammation biomarkers such as interleukin (IL)-6 and tumor necrosis factor (TNF)-α (two pro-inflammatory cytokines) were higher in individuals with OSA [11,12].

Another inflammatory biomarker is the high sensitivity C-reactive protein (hs-CRP). The expression of hs-CRP is IL-6 dependent in the kidney and largely under the IL-6 regulation [13,14]. Further, higher hs-CRP plasma levels predicted the risk of the cardiovascular disease (CVD), diabetes and impaired cognitive functions [14]. Likewise, elevated serum levels of hs-CRP and CRP are independent risk factors of CVD [15]. Unlike other cytokines, CRP levels can be quite stable over twenty-four hours and may reflect the inflammatory response level [16].

As mentioned above, OSA has a particularly close relationship with obesity [7], and up to 40% of the risk of OSA is genetically predisposed [17]. Further, obesity is associated with the condition of chronic low-grade inflammation, characterized by an increase in some inflammatory markers such as CRP [18].

Until now, there were four meta-analyses related to the association between OSA and CRP. The first meta-analysis in 2013 reported blood levels of CRP based on 25 studies [19]; the second meta-analysis in 2015 included 11 studies (5 related to CRP and 6 related to hs-CRP) [20]; the third meta-analysis in 2017 included 15 studies (7 related to CRP and 8 related to hs-CRP) [21], and fourth meta-analysis in 2019 reported five studies in adult non-smoking individuals with OSA [22]. The overall results were that CRP circulating levels in individuals with OSA were significantly higher, compared to healthy matched controls. Further, individuals with OSA, particularly those with moderate–severe OSA and BMI ≥ 30 kg/m^2^, had significantly elevated levels of CRP/hs-CRP. Next, levels of CRP were positively associated with the severity of OSA, and higher AHI indices were correlated with higher CRP levels. While these results were already of clinical and practical importance to increase the understanding of the underlying pathophysiology and above all inflammation-related mechanisms of OSA, the present meta-analysis expanded upon previous meta-analyses in four ways:We included 109 studies (96 in adults and 13 in children)We reported subgroup and meta-regression analyses in adults with OSA compared to controls on the serum and plasma levels of hs-CRPWe reported subgroup and meta-regression analyses in adults with OSA compared to controls on the serum and plasma levels of CRPWe reported serum and plasma levels of both hs-CRP and CRP in children with OSA, always compared to controls.

We claim that the present results may further improve the understanding of the inflammation-related physiological mechanisms both in children and adults with OSA. 

## 2. Materials and Methods

To report the results of the in systematic review, we followed the Preferred Reporting Items for Systematic Reviews and Meta-Analyses (PRISMA) guidelines [23].

### 2.1. Search Strategy and Study Selection

One author (M.S.) comprehensively searched PubMed/Medline, Scopus, Cochrane Library, and Web of Science databases to retrieve articles published until 31 May 2020, with no restrictions. The searched terms were (“obstructive sleep apnea” or “sleep apnea” or “OSA” or “obstructive sleep apnea syndrome” or “OSAS”) and (“C-reactive protein” or “CRP” or “high sensitivity C-reactive protein” or “hs-CRP”) and (“plasma” or “serum”). We manually searched the references (meta-analyses and original and review articles) related to our topics. The titles and abstracts of the retrieved studies were independently read by two authors (M.M.I. and F.F.). Then, the two authors (M.M.I. and F.F.) selected the relevant studies and another author (M.S) retrieved the full-texts of the articles.

### 2.2. Eligibility Criteria

Inclusion criteria were: 1. case-control studies evaluating the association between plasma or serum and hs-CRP and CRP levels and OSA without age, sex or BMI restrictions; 2. OSA was diagnosed as apnea-hypopnea index (AHI) > 5 events/h in adults, and AHI > 1 events/h in children; 3. OSA was determined with polysomnography; 4. There are no other systemic diseases such as diabetes, neurological disorders such as multiple sclerosis, neurodegenerative disorders such as Alzheimer’s disease in two groups (individuals with OSA and controls); 5. studies reporting pretreatment morning serum or plasma levels of hs-CRP and CRP (around 6–10 am); 6. studies reporting sufficient data to compute the mean difference (MD) and 95% confidence interval (CI) in two groups; 7. studies with more than 10 cases included as in two groups.

Exclusion criteria were: 1. studies with irrelevant or insufficient data or without clinical data; 2. meta-analyses, review articles, animal studies, book chapters, and conference papers; 3. studies without a control group; 4. studies reporting levels of hs-CRP and CRP in obstructive sleep apnea hypopnea syndrome (OSAHS) or sleep-disordered breathing (SDB); 5. studies reporting controls with AHI > 5 events/h in adults and AHI > 1 events/h in children; 5. studies with overlapped data with other studies already included in the present analysis.

### 2.3. Data Extraction

The data from each study included in the meta-analysis were independently extracted by two authors (S.B. and M.S). If there was a disagreement between both authors, the third author (M.M.I.) helped to make a final decision. The data included fundamental information involving the first author, publication year, country of study, ethnicity of participants in each study, age, BMI, and AHI of both groups, and mean and standard deviation (SD) of plasma and serum levels of CRP and hs-CRP.

### 2.4. Quality Assessment

The quality of the studies included in the analysis was evaluated by one author (M.S.) using the Newcastle-Ottawa Scale (NOS); nine was a maximum total score of each study [24].

### 2.5. Statistical Analyses

The data were analyzed by one author (M.S.) to calculate the raw MD and 95% CI. Review Manager 5.3 (RevMan 5.3) software was used that evaluated the significance of the pooled MD with Z test. Heterogeneity was examined during the studies using both Cochrane Q [25] and I^2^ metrics with scores ranging from 0 to 100% [26]. In addition, when there were heterogeneity values of (*P*_heterogeneity_ or *P*_h_ < 0.1) and I^2^ > 50%, this means a statistically significant heterogeneity, a random-effects model analysis was performed to evaluate the pooled MD and 95% CI values. Otherwise, we used fixed-effects model.

The results of the bias (Begg’s [27] and Egger’s [28] tests) were estimated by the Comprehensive Meta-Analysis version 2.0 (CMA 2.0) software. Subgroup analyses were performed based on ethnicity, AHI, BMI, and number of participants. The sensitivity analyses, namely the “cumulative analysis” and “one study removed”, were used to evaluate the consistency/stability of the results. If the *p*-value (2-tailed) was less than 0.05, there was a statistically significant difference.

The meta-regression as a quantitative method was used in meta-analyses to estimate the impact of moderator variables on study effect sizes based on the year of publication, the mean age, the mean BMI, the mean AHI, and number of participants.

The trim-and-fill method aims to estimate potentially missing studies due to publication bias in the funnel plot and to adjust the overall effect estimate [29].

Some studies reported values of CRP or hs-CRP in standard errors (SE); in this case, SEs were changed into standard deviation (SD), (SE = SD/√ N; N = number of individuals). Some studies reported “median and interquartile values” or “median and range”; in this case these indices were changed into “mean and SD” utilizing methods outlined by Hozo et al. [30,31].

The serum and plasma hs-CRP and CRP levels were reported in milligram per deciliter (mg/dL). Cases with a BMI more than 30 kg/m^2^ were considered obese [32]. For studies with data reported on the graphs, we used GetData Graph Digitizer 2.26 software.

## 3. Results

### 3.1. Study Selection

A total of 1046 records were retrieved from the databases (Figure 1). After removing duplicates, 558 records were screened; next, 339 records were excluded, and therefore 219 full text articles were evaluated for eligibility. Then, 110 full text articles were excluded for the following reasons (9 reviews. 4 meta-analyses. 39 articles had no control group or included control group with an AHI > 5 in adults. 3 articles included control group with an AHI > 1 in children. 1 article didn’t show AHI. 4 articles included groups under treatment. 4 articles included overlap data with other studies. 8 articles included OSA participations with other diseases. 17 articles included OSAS. 5 articles included less than 10 cases in one group or both groups (case or control groups). 3 letters to editor. 2 animal studies. 4 included participants with SDB. 2 articles didn’t include data. 1 book chapter. 4 articles reported irrelevant data). At last, 109 studies were selected for the analysis. 

### 3.2. Features of the Studies

The characteristics of the studies entered in the analysis are shown in Table 1. Of the 109 studies, 96 studies reported blood levels of hs-CRP/CRP in adults, and 13 studies reported blood levels of hs-CRP/CRP in children. The studies were published from 2002 to 2020. Of the 96 studies on blood levels of hs-CRP/CRP in adults, 11 studies reported plasma hs-CRP, 44 studies reported serum hs-CRP, 9 studies reported plasma CRP, and 32 studies reported serum CRP levels. Six out of 13 studies in children reported plasma hs-CRP, 4 studies reported serum hs-CRP, one study reported plasma CRP, and 2 studies reported serum CRP levels.

### 3.3. Plasma hs-CRP Levels in Adults with Obstructive Sleep Apnea

Eleven studies of plasma hs-CRP levels in adults involved 1365 individuals with OSA and 1629 controls (Table 2). The pooled MD of plasma hs-CRP levels in individuals with OSA in comparison to controls was 0.11 mg/dL [95% CI: 0.07, 0.16; *p* < 0.00001; I^2^ = 89% (*P*_heterogeneity_ or *p*_h_ < 0.00001)]. Thus, plasma hs-CRP levels were significantly higher in individuals with OSA than in controls.

### 3.4. Serum hs-CRP Levels in Adults with Obstructive Sleep Apnea

Forty-four studies of serum hs-CRP levels in adults involved 3857 individuals with OSA and 1240 controls (Table 3). The pooled MD of serum hs-CRP levels in individuals with OSA in comparison to controls was 0.09 mg/dL [95% CI: 0.07, 0.11; *p* < 0.00001; I^2^ = 96% (*p*_h_ < 0.00001)]. Thus, serum hs-CRP levels were significantly higher in individuals with OSA than in controls.

### 3.5. Plasma CRP Levels in Adults with Obstructive Sleep Apnea

Nine studies of plasma CRP levels in adults involved 592 individuals with OSA and 346 controls (Table 4). The pooled MD of plasma CRP levels in individuals with OSA in comparison to controls was 0.06 mg/dL [95% CI: −0.24, 0.36; *p* = 0.72; I^2^ = 99% (*p*_h_ < 0.00001)]. Thus, plasma CRP levels had no significant difference in individuals with OSA compared to controls.

### 3.6. Serum CRP Levels in Adults with Obstructive Sleep Apnea

Thirty-two studies of serum CRP levels in adults involved 2562 individuals with OSA and 1315 controls (Table 5). The pooled MD of serum CRP levels in individuals with OSA in comparison to controls was 0.36 mg/dL [95% CI: 0.28, 0.45; *p* < 0.00001; I^2^ = 96% (*p*_h_ < 0.00001)]. Thus, serum CRP levels were significantly higher in individuals with OSA than in controls.

### 3.7. Plasma and Serum Levels of hs-CRP and CRP in Children with Obstructive Sleep Apnea

Six studies of plasma hs-CRP levels involved 600 children with OSA and 704 controls, four studies of serum hs-CRP levels involved 177 individuals with OSA and 95 controls, one study of plasma CRP levels involved 84 individuals with OSA and 22 controls, and two studies of serum CRP levels involved 93 individuals with OSA and 69 controls, in children (Table 6). The pooled MD of plasma hs-CRP, serum hs-CRP, plasma hs-CRP, and serum hs-CRP in individuals with OSA in comparison to controls was 1.17 mg/dL [95% CI: 0.35, 1.98; *p* = 0.005; I^2^ = 94% (*p*_h_ < 0.00001)], 0.18 mg/dL [95% CI: −0.00, 0.35; *p* = 0.05; I^2^ = 98% (*p*_h_ < 0.00001)], 0.08 mg/dL [95% CI: − 0.02, 0.18; *p* = 0.10], and 0.04 mg/dL [95% CI: −0.04, 0.13; *p* = 0.33; I^2^ = 0% (*p*_h_ = 0.36)], respectively. Thus, just plasma hs-CRP levels were significantly higher in individuals with OSA than in controls.

### 3.8. Subgroup Analysis of Blood hs-CRP Levels in Adults with Obstructive Sleep Apnea

#### 3.8.1. Ethnicity

Subgroup analyses of plasma and serum hs-CRP levels in adults are reported in Table 7. The pooled analysis showed that for those with OSA, plasma hs-CRP levels in Caucasian (MD = 0.10 mg/dL, *p* = 0.004) and Asian (MD = 0.12 mg/dL, *p* = 0.003) ethnicities were significantly higher than the plasma hs-CRP levels of the respective controls, not for and mixed ethnicity (MD = 0.24 mg/dL, *p* = 0.47). In addition, the pooled analysis showed that for those with OSA, serum hs-CRP levels in Caucasian (MD = 0.18 mg/dL, *p* < 0.00001) and Asian (MD = 0.08 mg/dL, *p* < 0.00001) ethnicities were significantly higher than the serum hs-CRP levels of the respective controls, not for mixed ethnicity (MD = 0.05 mg/dL, *p* = 0.28). Therefore, ethnicity could be an effective factor on the serum and plasma levels of hs-CRP as the significant difference was found for Asian and Caucasian ethnicities, not for mixed ethnicity. 

#### 3.8.2. Mean BMI of Participants

With regards to mean BMI of participants with OSA, the pooled MD of plasma hs-CRP levels of individuals with OSA was significantly higher than in controls, irrespective of their BMI: mean BMI ≤ 30 kg/m^2^ (MD = 0.11 mg/dL, *p* = 0.003), but there was no for mean BMI > 30 kg/m^2^ (MD = 0.10 mg/dL, *p* = 0.15). The pooled MD of serum hs-CRP levels of individuals with OSA was significantly higher than in controls, irrespective of their BMI: mean BMI > 30 kg/m^2^ (MD = 0.18 mg/dL, *p* < 0.0001); or mean BMI ≤ 30 kg/m^2^ (MD = 0.08 mg/dL, *p* < 0.00001). With regards to mean BMI of controls, the pooled MD of plasma hs-CRP levels was significantly higher in individuals with OSA compared to controls, irrespective of whether the BMI of controls was ≤30 kg/m^2^ (MD = 0.11 mg/dL, *p* < 0.001), but there was no significant difference for mean BMI > 30 kg/m^2^ (MD = 0.15 mg/dL, *p* = 0.30). The pooled MD of serum hs-CRP levels of individuals with OSA was significantly higher than in controls, irrespective of whether the BMI of controls was >30: mean BMI > 30 kg/m^2^ (MD = 0.11 mg/dL, *p* = 0.004); or mean BMI ≤ 30 kg/m^2^ (MD = 0.09 mg/dL, *p* < 0.00001). Therefore, BMI could be an effective factor on the plasma level of hs-CRP as the significant difference was found for participants with BMI ≤ 30 kg/m^2^, not for BMI > 30 kg/m^2^. However, BMI couldn’t be an effective factor on the serum level of hs-CRP.

#### 3.8.3. Total Number of Participants

With regard to the number of participants in each study, the pooled MD of plasma hs-CRP levels in those studies with more than 100 cases across OSA and control groups was 0.14 mg/dL (*p* = 0.04); the pooled MD of the studies with ≤100 cases across the two groups was 0.11 mg/dL (*p* < 0.0002). In contrast, the MD of serum hs-CRP levels in studies with more than 100 cases was 0.10 mg/dL (*p* < 0.00001), and in studies with less than 100 cases across two groups was 0.08 mg/dL (*p* < 0.00001). Therefore, number of participants couldn’t be an effective factor on the plasma and serum levels of hs-CRP.

#### 3.8.4. Mean AHI of Participants with Obstructive Sleep Apnea

With regard to the mean AHI of participants with OSA, the pooled MD of plasma hs-CRP levels in the studies including a mean AHI > 30 events/h was 0.14 mg/dL (*p* = 0.02); the pooled MD of the studies with a mean AHI ≤ 30 events/h was 0.10 mg/dL (*p* = 0.01). In contrast, the MD of serum hs-CRP levels in the studies including a mean AHI > 30 events/h was 0.11 mg/dL (*p* < 0.00001) and in the studies including a mean AHI ≤ 30 events/h was 0.07 mg/dL (*p* < 0.00001). Therefore, AHI couldn’t be an effective factor on the plasma and serum levels of hs-CRP.

### 3.9. Subgroup Analysis of Blood CRP Levels in Adults with Obstructive Sleep Apnea

#### 3.9.1. Ethnicity

Subgroup analyses of plasma and serum CRP levels in adults are reported in Table 8. The pooled analysis showed that for those with OSA, there was no significant difference for plasma CRP levels in Caucasian (MD = 0.22 mg/dL, *p* = 0.39) and Asian (MD = 0.21 mg/dL, *p* = 0.18) ethnicities, compared to the respective controls. In addition, the pooled analysis showed that for those with OSA, serum CRP levels in Caucasian (MD = 0.38 mg/dL, *p* < 0.00001) and Asian (MD = 0.38 mg/dL, *p* < 0.00001) ethnicities were significantly higher than serum CRP levels of the respective controls, not for mixed ethnicity (MD = 0.06 mg/dL, *p* = 0.37). Therefore, ethnicity couldn’t be an effective factor on the plasma and serum levels of CRP.

#### 3.9.2. Mean BMI

With regards to mean BMI of participants with OSA, the pooled MD of plasma CRP levels had no significant difference in individuals with OSA compared to controls, irrespective of their BMI: mean BMI ≤ 30 kg/m^2^ (MD = 0.95 mg/dL, *p* = 0.001). In contrast, for mean BMI > 30 kg/m^2^ (MD = −0.088 mg/dL, *p* = 0.04) pooled MD of plasma CRP levels was significantly lower than in controls. The pooled MD of serum CRP levels of individuals with OSA was significantly higher than in controls, irrespective of their BMI: mean BMI > 30 kg/m^2^ (MD = 0.31 mg/dL, *p* < 0.00001); or mean BMI ≤ 30 kg/m^2^ (MD = 0.41 mg/dL, *p* < 0.00001). With regards to mean BMI of controls, there was no significant difference for the pooled MD of plasma CRP levels in individuals with OSA, compared to controls, whether the BMI of controls was ≤30 kg/m^2^ (MD = 0.06 mg/dL, *p* = 0.72); further, there was no study with individuals with a mean BMI > 30 kg/m^2^. The pooled MD of serum CRP levels of individuals with OSA was significantly higher than in controls in those studies including a mean BMI > 30 kg/m^2^ (MD = 0.61 mg/dL, *p* = 0.002); or mean BMI ≤ 30 kg/m^2^ (MD = 0.35 mg/dL, *p* < 0.00001) in controls. Therefore, BMI couldn’t be an effective factor on serum levels of CRP.

#### 3.9.3. Total Number of Participants

With regard to the number of participants in each study, the pooled MD of plasma CRP levels in the studies including ≤100 cases across OSA and control groups was 0.69 mg/dL (*p* = 0.0003), and there was no significant difference for the studies including more than 100 cases across the two groups (MD = −1.43 mg/dL (*p* = 0.14). In contrast, the MD of serum CRP levels in the studies with >100 cases was 0.21 mg/dL (*p* = 0.0003) and in the studies including ≤100 cases across two groups was 0.53 mg/dL (*p* < 0.00001). Therefore, the number of participants could be an affective factor on plasma levels of CRP as the significant difference was found for the studies with ≤100 cases, not for >100 cases. However, the number of participants couldn’t be an affective factor on serum levels of CRP.

#### 3.9.4. Mean AHI of Participants with Obstructive Sleep Apnea

With regard to the mean AHI of participants with OSA, the pooled MD of plasma CRP levels in the studies including a mean AHI ≤ 30 events/h was 0.18 mg/dL (*p* = 0.02), and there was no significant difference for the studies with a mean AHI ≤ 30 events/h (MD = 1.64 mg/dL (*p* = 0.16). In contrast, the MD of serum CRP levels in the studies including a mean AHI > 30 events/h was 0.54 mg/dL (*p* < 0.00001) and in the studies including a mean AHI ≤ 30 events/h was 0.27 mg/dL (*p* < 0.00001). Therefore, AHI could be an affective factor on plasma levels of CRP as the significant difference was found for the studies with AHI ≤ 30 events/h, not for AHI > 30 events/h. However, AHI couldn’t be an affective factor on serum levels of CRP.

### 3.10. Meta-Regression Analysis of Blood hs-CRP Levels in Adults with Obstructive Sleep Apnea 

The results of meta-regression of plasma and serum hs-CRP levels are shown in Table 9. The year of publication, mean age, mean BMI, and number of participants had no independent significant effects on serum or plasma hs-CRP levels, whereas with greater AHI, serum hs-CRP levels were significantly higher.

### 3.11. Meta-Regression Analysis of Blood CRP Levels in Adults with Obstructive Sleep Apnea 

The results of meta-regression of plasma and serum CRP levels are shown in Table 10. The year of publication, mean age, mean BMI, mean AHI, and number of participants had no independent significant effects on serum or plasma CRP levels.

### 3.12. Quality Scores

The quality scores of the studies included in the analysis are shown in Table 11.

### 3.13. Sensitivity Analysis

The “cumulative analysis” and the “one study removed” as two sensitivity analyses revealed the stability of the results in adults and in children.

### 3.14. Publication Bias of Blood hs-CRP and CRP Levels in Adults with Obstructive Sleep Apnea

The funnel plots of the analysis of plasma and serum levels of CRP and hs-CRP are shown in Figure 2, and Table 12 shows the results of the trim-and-fill method on bias.

For plasma and serum hs-CRP levels, Egger’s test (*p* = 0.86142 and *p* = 0.06867, respectively) and Begg’s test (*p* = 0.93795 and *p* = 0.10132, respectively) indicated no bias either between or across the studies.

For plasma hs-CRP levels without imputed studies, under the fixed-effects model, the point estimate and 95% CI for the combined studies is 0.114 (0.097, 0.130); using the trim–fill method, the imputed point estimate is similar. In addition, under the random-effects model, the point estimate and 95% CI for the combined studies is 0.115 (0.064, 0.0166); using the trim–fill method, the imputed point estimate is similar. For serum hs-CRP levels and 14 imputed studies, under the fixed-effects model, the point estimate and 95% CI for the combined studies is 0.049 (0.047, 0.052), and using trim–fill method, the imputed point estimate is 0.049 (0.046, 0.051). In addition, under the random-effects model, the point estimate and 95% CI for the combined studies is 0.089 (0.069, 0.109); using the trim–fill method, the imputed point estimate is 0.059 (0.038, 0.080). These findings showed that the overall effect sizes for serum and plasma hs-CRP levels reported in the forest plot appeared to be valid, with a trivial publication bias effect based on fixed-effects or random-effects models, because the observed estimates were similar to the adjusted estimates.

For plasma and serum CRP levels, Egger’s test (*p* = 0.98284 and *p* = 0.00341, respectively) revealed a bias for serum level, but not for plasma level, and Begg’s test (*p* = 0.53161 and *p* = 0.16860, respectively) indicated no bias either between or across the studies for both samples.

For plasma CRP levels and one imputed study, under the fixed-effects model, the point estimate and 95% CI for the combined studies was 0.131 (0.096, 0.165); using the trim–fill method, the imputed point estimate was 0.125 (0.91, 0.160). In addition, under the random-effects model, the point estimate and 95% CI for the combined studies was 0.107 (−0.256, 0.471); using the trim–fill method, the imputed point estimate was −0.317 (−0.721, 0.087). For serum CRP levels and 13 imputed studies, under the fixed-effects model, the point estimate and 95% CI for the combined studies was 0.094 (0.079, 0.108), and using trim–fill method, the imputed point estimate was 0.55 (0.41, 0.69). In addition, under the random-effects model, the point estimate and 95% CI for the combined studies was 0.304 (0.216, 0.391); using the trim–fill method, the imputed point estimate was 0.083 (−0.011, 0.177). These findings showed that the overall effect sizes for CRP levels reported in the forest plot appeared to be valid, with a trivial publication bias effect based on fixed-effects model for plasma level, because the observed estimates were similar to the adjusted estimates. In contrast, the overall effect sizes on CRP levels reported in the forest plot appeared to be invalid, with a significant publication bias effect based on random-effects model for both samples (serum and plasma levels) and fixed-effects model for serum level, because the observed estimates were substantially different to the adjusted estimates.

## 4. Discussion

The main findings of the present systematic review, meta-analysis and meta-regression on the plasma and serum levels of hs-CRP and CRP in adults and children were as follows:serum and plasma hs-CRP levels and serum CRP levels in adults were significantly higher in individuals with OSA than in controls.there was no significant difference in adults with OSA compared to controls for plasma levels of CRP.in children, just plasma hs-CRP levels were significantly higher in pediatric individuals with OSA, compared to controls.based on subgroup analysis for the plasma and serum levels of hs-CRP, ethnicity and mean BMI in individuals with OSA could impact on the results of plasma hs-CRP levels, and ethnicity on serum levels of hs-CRP.based on subgroup analysis for the plasma and serum levels of CRP, number of participants and mean AHI in individuals with OSA could impact on plasma CRP levels, and ethnicity on serum levels of CRP.based on meta-regression on the plasma and serum levels of hs-CRP and CRP, just mean AHI of individuals with OSA could be an interfering factor on the results of serum levels of hs-CRP.

It follows that with the present meta-analysis, systematic review and meta-regression, we were able to show that inflammatory markers, here hs-CRP and CRP, are increased in individuals with OSA, though, the pattern of results is not straightforward and uniform; rather, hs-CRP and CRP levels might be higher compared to, or equal to hs-CRP and CRP levels of healthy controls, depending of the blood samples (serum vs. plasma), age (children vs. adults), ethnicity, BMI, sample size and AHI-cut-off values. Given this, it seems conceivable that previous and present results on the associations between hs-CRP and CRP in individuals with OSA appear inconsistent. 

Indeed, there were conflicting results for the associations between OSA and hs-CRP [33,52,59,60,80,104] or CRP [36,57,61,65] levels. One meta-analysis [19] included 25 studies reporting the blood levels of CRP in individuals with and without OSA. Its results showed that the pooled standardized mean difference in adults with OSA compared to controls was 1.77 (*p* < 0.0001). Three meta-analyses [20,21,22] showed serum levels of hsCRP and/or CRP elevated in individuals with OSA compared to controls. The results were in line with our meta-analysis for serum levels, but not plasma levels. Therefore, the type of sample for measuring CRP appears to be important in adults with OSA; however, there was no difference between both samples (plasma or serum) for measuring hs-CRP in our meta-analysis. The meta-regression in the meta-analysis [19] showed significant effects of age, BMI, and AHI on CRP levels similar to the results of the study of Kanbay et al. [140]. In contrast, the meta-regression didn’t show the effect of these factors on serum or plasma levels of CRP; rather, just AHI had a significant effect on serum levels of hs-CRP. Further, in line with two studies [43,46], age was not associated with CRP levels. To explain these conflicting results, it is conceivable that a higher number of studies and the separate analysis of both serum and plasma yielded these differences between the present and former analyses.

For OSA, the following risk factors were reported: Older age, male gender, obesity, heavy drinking, smoking, and anatomical abnormalities, though, obesity appeared to be the strongest risk factor [141]. CRP is a sensitive marker for systemic inflammation, with increased plasma levels in this acute phase reactor indicating increased inflammatory activity in humans. Further, CRP can also be a substitute biomarker of low-grade inflammation related to obesity [58]: Studies [36,42,46] reported that CRP levels were significantly higher in obese individuals with OSA, compared to non-obese individuals with OSA. In contrast, Jung et al. [115] showed that there was no association between BMI and hs-CRP levels and in our meta-analysis, the mean BMI couldn’t be an effective factor on the plasma and serum levels of CRP and hs-CRP.

There were relationships between OSA severity and CRP [33,35,64,92] and hs-CRP [100,106,123] levels. One study [142] showed that hs-CRP levels were related to OSA independently of visceral obesity. Jin et al. [46] reported that serum CRP levels positively correlated with AHI. In our meta-analysis, mean AHI couldn’t be an affective factor on serum levels of CRP. In our subgroup analyses, we found that hs-CRP and CRP levels in the studies with a mean AHI ≤ 30 events/h were higher than in those studies with a mean AHI > 30 events/h. Further, AHI was just a confounding factor for serum hs-CRP levels and our AHI couldn’t be an affective factor on the plasma and serum levels of CRP and hs-CRP.

In children, there was just the association between the plasma hs-CRP level and OSA with a higher level in the participants with OSA compared to the controls. A lack of association for the serum hs-CRP, plasma CRP, and serum CRP could be due to few include studies.

To explain the associations between hs-CRP, plasma CRP and serum CRP, the following assumptions were made: First, C-reactive protein (CRP) is regarded as a significant serum marker of inflammation; as such, CRP is synthesized in the liver, mainly regulated via the expression of IL-6 [13]. IL-6 concentrations are increased in individuals with OSA(S), compared to healthy controls [12]. As such, it appears plausible that higher IL-6 concentrations are associated with higher h-CRP, plasma and serum CRP concentrations. Second, higher CRP concentrations were observed among individuals at increased risk of cardiovascular diseases, such as atherosclerosis, stroke, and myocardial infarction [100,143]. As such, it appeared conceivable that increased serum CRP levels could predict complications such as atherosclerosis [100,143]. In this view, long-term sustained hypoxia could yield activated inflammatory responses with elevated levels of proinflammatory cytokines [144].

The novelty of the results should be balanced against the following limitations: First, in all studies the results were not adjusted based on possible confounding factors such as obesity, smoking, and alcohol consumption, and even in some studies based on gender. Second, there was a high heterogeneity and also a bias between and across studies, all of which could impact on the reliability of the results. Third, studies with a small sample size (less than 100) might have had insufficient power to detect associations between OSA, CRP and hs-CRP levels. Fourth, studies reported various cut-off AHI values, which made it difficult to compare the study results. Fifth, in some studies, CRP and hs-CRP levels were considered as a secondary outcome.

By contrast, the meta-analysis had several strengths: First, the meta-regression showed that higher AHI could be a cofounder factor on serum hs-CRP levels. Second, there were sufficient studies to allow the subgroup analysis. Third, sensitivity analysis illustrated the consistency of results. Fourth, we included the studies written in any language. Fifth, unlike previous meta-analyses, we included also data of children with OSA.

Last, while we focused on significant *p*-values, we did less rely on effect size calculation. However, a closer inspection showed that when focusing on effect sizes, the overall pattern of results did not change.

## 5. Conclusions

The findings of this systematic review, meta-analysis and meta-regression demonstrated that higher plasma and serum levels of hs-CRP and serum levels of CRP in individuals with OSA appeared to be associated with the disease severity. Further, higher AHI appeared to impact on the associations between OSA and CRP/hs-CRP levels. Future studies might consider, if and to what extent interventions on OSA (e.g., using CPAP devices) may favorably impact on CRP/hs-CRP levels and possibly also on weight regulation. While it appeared that the use of CPAP did not change the level of inflammatory markers after 12 weeks among women with Obstructive Sleep Apnea [145], it would be of clinical and practical importance to know if changes is hs-CRP and CRP would be associated with weight loss among individuals with OSA.

We claim that the present results are of clinical importance: Individuals with OSA appeared to be at increased risk to suffer from higher hs-CRP and CRP levels; such increased inflammatory levels were associated with a higher risk of suffering from further diseases such as early neurological deterioration following stroke [146] or major cardiovascular events in patients with peripheral artery disease [147]. We hold that the present results are also of practical importance, because individuals with higher hs-CRP or CRP levels appeared to be at a dramatically increased risk of reporting a much severe progress of the COVID-19 [148,149]; indeed; higher CRP levels were an independent factor to predict the severity of COVID-19 [149]; given this, we speculate that individuals with OSA and higher hs-CRP and CRP levels might also be at an increased risk to suffer from severe COVID-19.

## Figures and Tables

**Figure 1 life-11-00362-f001:**
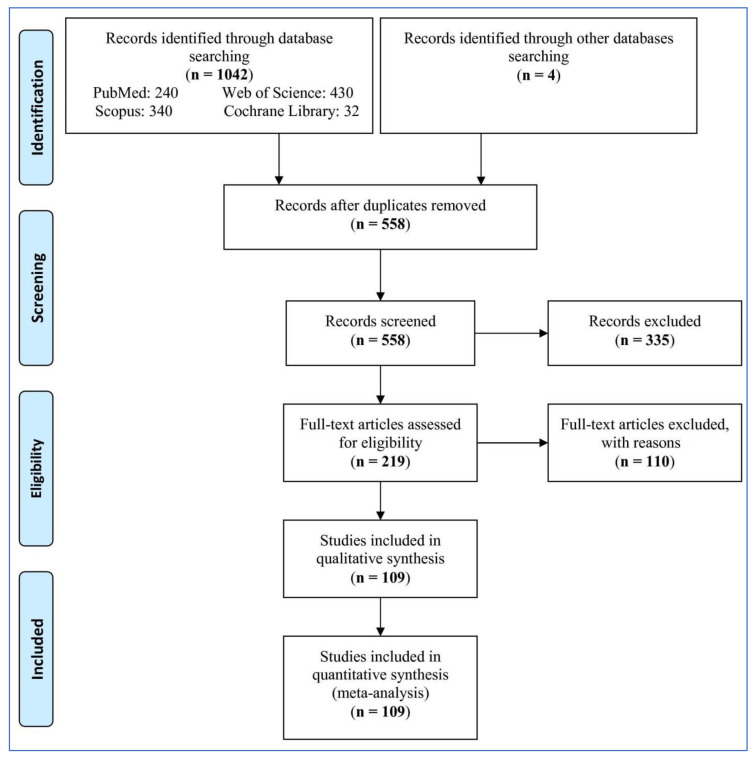
Flowchart of the study selection.

**Figure 2 life-11-00362-f002:**
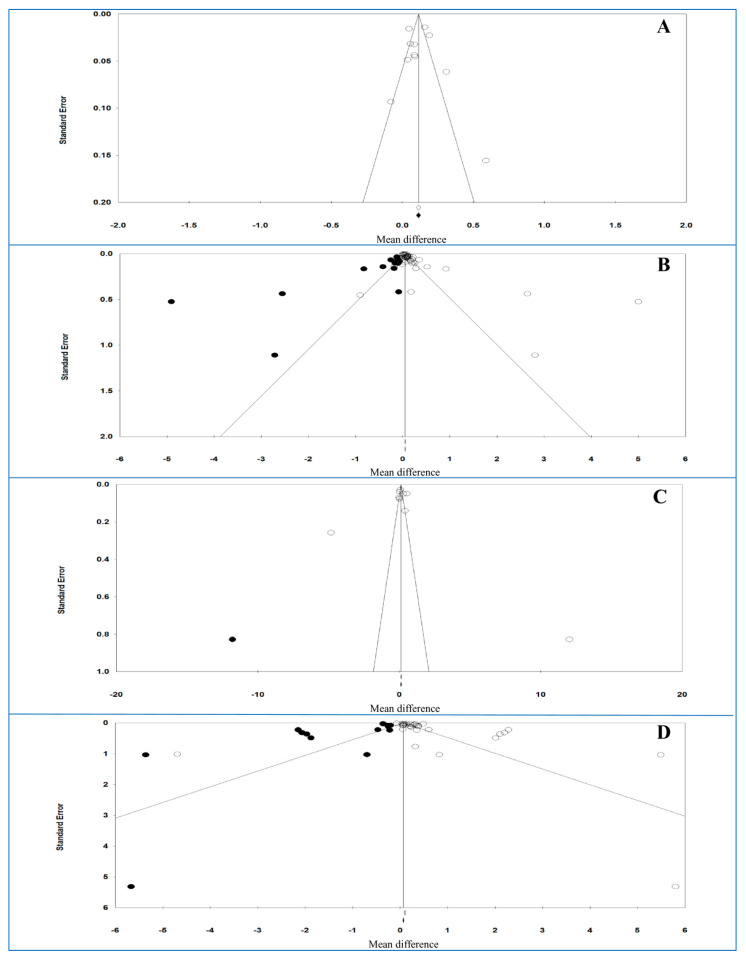
Funnel plot of analysis of high-sensitivity C-reactive protein (hs-CRP) (**A**: plasma and **B**: serum) and CRP levels (**C**: plasma and **D**: serum) in adult participants. Open circles represent observed studies. Black circles represent imputed studies. Open diamonds represent the pooled effects from the original studies. Black diamonds represent the pooled effects incorporating the imputed studies.

**Table 1 life-11-00362-t001:** Characteristics of studies included in the meta-analysis (n = 109).

First Author, Year	Country	Ethnicity	No. of OSA /Control	OSA Patients			Controls			Biomarker	Sample
				Age (Years)	BMI (kg/m^2^)	AHI (Events/h)	Age (Years)	BMI (kg/m^2^)	AHI (Events/h)		
Adults											
Shamsuzzaman, 2002 [33]	USA	Mixed	22/20	48 ± 14.07	36 ± 18.76	60 ± 23.45	43 ± 13.41	34 ± 17.88	3 ± 4.47	hs-CRP	Plasma
Teramoto, 2003 [34]	Japan	Asian	40/40	Adult	NR	≥5	Adult	NR	<5	hs-CRP	Plasma
Yokoe, 2003 [35]	Japan	Asian	26/14	52.5 ± 29.07	28.4 ± 17.85	33.7 ± 12.75	48.8 ± 11.72	27.6 ± 1.87	2.8 ± 0.74	hs-CRP	Serum
Barceló, 2004 [36]	Spain	Caucasian	47/18	48.5 ± 9	30.4 ± 2.5	46 ± 11	47 ± 5	25.5 ± 2.9	2 ± 1	CRP	Plasma
Guilleminault, 2004 [37]	USA	Mixed	146/54	46.81 ± 11.42	29.39 ± 7.05	≥5	43.87 ± 9.79	24.74 ± 5.34	<5	CRP	Serum
Minoguchi, 2005 [38]	Japan	Asian	36/16	48.22 ± 14.1	28.01 ± 4.44	34.89 ± 17.46	46.5 ± 15.2	28.3 ± 5.2	3.3 ± 3.6	CRP	Serum
Can, 2006 [39]	Turkey	Caucasian	62/30	47.14 ± 1.62	29.63 ± 0.67	25.04 ± 3.85	43.5 ± 2.1	26.0 ± 0.78	<5	CRP	Serum
Minoguchi, 2006 [40]	Japan	Asian	40/30	49.8 ± 18.32	28.05 ± 04.74	28.15 ± 37.28	47.99 ± 12.58	25.75 ± 3.82	3.76 ± 2.16	hs-CRP	Serum
Shiina, 2006 [41]	Japan	Asian	94/90	52 ± 9.69	28.1 ± 4.84	47.2 ± 19.38	47 ± 9.48	25.7 ± 3.79	4.2 ± 3.79	CRP	Plasma
Ryan, 2007 [42]	Ireland	Caucasian	66/30	42.5 ± 8.5	32.5 ± 4.8	35.0 ± 13.9	41 ± 8	30.7 ± 3.1	1.2 ± 1.0	hs-CRP	Serum
Chung, 2007 [43]	Korea	Asian	68/22	42.72 ± 9.12	26.47 ± 3.22	≥5	42.1 ± 8.7	26.2 ± 3.9	<5	hs-CRP	Plasma
Iesato, 2007 [44]	Japan	Asian	155/39	49.8 ± 1.1	28.9 ± 0.4	36.7 ± 2.1	47.7 ± 2.2	25.6 ± 0.6	1.8 ± 0.2	hs-CRP	Serum
Minoguchi, 2007 [45]	Japan	Asian	50/15	49 ± 12.72	27.5 ± 3.88	27.21 ± 15.41	48.5 ± 11.99	28.1 ± 3.87	3.1 ± 1.54	hs-CRP	Serum
Jin, 2007 [46]	China	Asian	51/25	49.4 ± 10.5	27.5 ± 1.2	37.03 ± 4.81	49.9 ± 11.7	27.7 ± 0.9	2.9 ± 0.5	CRP	Serum
Kapsimalis, 2008 [47]	Greece	Caucasian	52/15	52.9 ± 12.7	30 ± 3.6	32.15 ± 10.4	47.0 ± 12.5	28.7 ± 4.3	3.1 ± 1.1	CRP	Serum
Saletu, 2008 [48]	Austria	Caucasian	103/44	55 ± 10	31 ± 5	37.62 ± 14.06	50 ± 14	27 ± 6	1.9 ± 1.3	hs-CRP	Serum
Sharma, 2008 [49]	India	Caucasian	29/68	45.28 ± 8.59	29.17 ± 4.02	48.64 ± 26.02	40.7 ± 9.3	26.72 ± 2.9	0.55 ± 1.24	hs-CRP	Serum
Takahashi, 2008 [50]	Japan	Asian	41/12	49.8 ± 10	29.4 ± 4.2	≥5	46.7 ± 11.2	25.7 ± 4.1	<5	hs-CRP	Plasma
Bhushan, 2009 [51]	India	Caucasian	62/46	43.8 ± 11.2	30.9 ± 4.4	≥5	41.7 ± 6.9	29.9 ± 3.0	<5	hs-CRP	Serum
Carneiro, 2009 [52]	Brazil	Mixed	16/13	40.1 ± 211.2	46.9 ± 8.0	65.7 ± 39.96	38.8 ± 11.88	42.8 ± 4.68	3.2 ± 01.81	hs-CRP	Plasma
Cofta, 2009 [53]	Poland	Caucasian	40/14	50.66 ± 10.33	30.13 ± 4.23	25.71 ± 8.04	50 ± 10	30.2 ± 5.4	2.2 ± 1.2	hs-CRP	Serum
Makino, 2009 [54]	Japan	Asian	157/24	52.59 ± 1.76	26.35 ± 0.5	28.77 ± 1.81	47.6 ± 2.8	25.0 ± 0.6	3.1 ± 0.4	CRP	Plasma
Sahlman, 2010 [55]	Finland	Caucasian	84/40	50.4 ± 9.3	32.5 ± 3.3	9.6 ± 2.9	45.6 ± 11.5	31.5 ± 3.5	1.9 ± 1.4	hs-CRP	Plasma
Aihara, 2011 [56]	Japan	Asian	150/20	57.0 ± 14.4	27.13 ± 5.49	36.44 ± 12.52	43.6 ± 17.7	24.8 ± 3.4	2.2 ± 1.5	hs-CRP	Serum
Barceló, 2011 [57]	Spain	Caucasian	119/119	46.0 ± 12.0	28.0 ± 4.0	38.56 ± 22.44	45.0 ± 11.0	28.0 ± 4.0	3.16 ± 2.0	CRP	Plasma
Basoglu, 2011 [58]	Turkey	Caucasian	36/34	50.0 ± 19.7	33.5 ± 5.7	≥5	49.7 ± 11.1	34.5 ± 2.9	<5	hs-CRP	Serum
Fredheim, 2011 [59]	Norway	Caucasian	84/53	48 ± 9.8	47.2 ± 6.1	≥5	36 ± 8.8	46.3 ± 5.2	<5	hs-CRP	Serum
Guasti, 2011 [60]	Italy	Caucasian	16/11	61 ± 10	31.76 ± 4.39	39.6 ± 19.1	55 ± 14	31.71 ± 4.44	<5	hs-CRP	Serum
Kanbay, 2011 [61]	Turkey	Caucasian	144/22	54.85 ± 11.82	33.03 ± 6.2	≥5	50.7 ± 13.9	29.3 ± 8.5	<5	CRP	Serum
Kasai, 2011 [62]	Japan	Asian	50/25	51 ± 11.8	17.3 ± 3.6	32.28 ± 13.04	50.9 ± 12.4	26.8 ± 3.7	2.7 ± 1.3	hs-CRP	Serum
Balci, 2012 [63]	Turkey	Caucasian	61/33	44.2 ± 10.8	27.2 ± 2.36	40.3 ± 27.3	41.6 ± 11.6	26.3 ± 1.4	3.2 ± 1.9	hs-CRP	Serum
Chien, 2012 [64]	Taiwan	Asian	30/30	50.5 ± 5.7	26.54 ± 2.40	48.4 ± 17.3	49.9 ± 6.8	25.87 ± 2.59	2.7 ± 1.3	hs-CRP	Serum
Feng, 2012 [65]	China	Asian	132/108	47.51 ± 10.31	27.17 ± 3.77	≥5	47.29 ± 10.89	27.07 ± 3.10	<5	CRP	Serum
Fornadi, 2012 [66]	Canada	Mixed	25/75	54.0 ± 12.0	29.0 ± 5.0	≥5	50.0 ± 13.0	26.0 ± 5.0	<5	CRP	Serum
Guven, 2012 [67]	Turkey	Caucasian	47/29	52.43 ± 8.19	29.62 ± 4.50	20.78 ± 3.03	53.24 ± 9.41	28.14 ± 3.77	<5	hs-CRP	Serum
Panoutsopoulos, 2012 [68]	Greece	Caucasian	20/18	54.30 ± 10.84	31.30 ± 2.00	25.35 ± 15.05	48.33 ± 7.67	30.00 ± 2.14	2.67 ± 1.41	CRP	Plasma
Chen, 2013 [69]	Taiwan	Asian	44/20	42.38 ± 11.95	27.11 ± 3.53	14.57 ± 2.85	42 ± 11	42 ± 11	3.3 ± 0.9	hs-CRP	Plasma
Kosacka, 2013 [70]	Poland	Caucasian	137/42	54.37 ± 9.83	34.28 ± 7.91	34.17 ± 21.77	50.69 ± 12.27	30.04 ± 5.40	2.14 ± 1.92	CRP	Serum
Wang, 2013 [71]	China	Asian	192/144	49.24 ± 9.93	26.89 ± 3.58	24 ± 8.89	48.74 ± 10.62	27.14 ± 3.28	2 ± 1.48	CRP	Serum
Zhang, 2013 [72]	China	Asian	75/23	32.2 ± 5.5	28.38 ± 3.56	26.65 ± 7.47	33.52 ± 4.71	26.42 ± 3.10	3.27 ± 1.62	hs-CRP	Serum
Akilli, 2014 [73]	Turkey	Caucasian	149/50	51.0 ± 9.1	32.0 ± 5.0	≥5	49.1 ± 8.5	29.6 ± 3.92	<5	hs-CRP	Serum
Ciccone, 2014 [74]	Italy	Caucasian	80/40	52.8 ± 10.6	28.6 ± 3.0	33.9 ± 21	52.3 ± 10.5	28.2 ± 2.7	2.1 ± 1.1	hs-CRP	Plasma
Li, 2014 [75]	China	Asian	156/110	47.0 ± 9.8	26.80 ± 3.2	23.71 ± 4.31	49.01 ± 8.11	27.36 ± 3.36	2.0 ± 1.48	CRP	Serum
Niżankowska-Jędrzejczyk, 2014 [76]	Canada	Caucasian	22/16	52.50 ± 8.33	30.15 ± 2.77	23.65 ± 11.51	54.06 ± 12.09	28.02 ± 3.36	2.24 ± 1.79	CRP	Plasma
Shi, 2014 [77]	China	Asian	126/74	48.67 ± 9.18	26.47 ± 2.38	≥5	49.15 ± 14.25	26.48 ± 2.41	<5	hs-CRP	Serum
Sökücü, 2014 [78]	Turkey	Caucasian	36/22	47.44 ± 11.68	33.10 ± 4.35	59.25 ± 18.99	40.76 ± 11.62	28.68 ± 6.09	3.41 ± 1.19	CRP	Serum
Yadav, 2014 [79]	UK	Caucasian	20/21	49 ± 10	52 ± 6	26.83 ± 23.85	45 ± 9	50 ± 8	4.93 ± 2.29	hs-CRP	Serum
Yüksel, 2014 [80]	Turkey	Caucasian	51/15	49 ± 10	31.0 ± 5.4	55.1 ± 17.2	46 ± 14	27.7 ± 3.9	1.5 ± 1.7	hs-CRP	Serum
Abakay, 2015 [81]	Turkey	Caucasian	44/49	47.4 ± 7.2	28.1 ± 6.3	25.1 ± 20.8	44.9 ± 11	25.8 ± 6.7	2.0 ± 0.9	CRP	Serum
Andaku, 2015 [82]	Brazil	Mixed	14/11	42.36 ± 9.48	26.65 ± 2.38	29.48 ± 22.83	43.00 ± 10.56	24.14 ± 2.67	2.71 ± 1.48	hs-CRP	Serum
da Silva Araújo, 2015 [83]	Brazil	Mixed	33/20	39.60 ± 1.48	34.39 ± 0.51	20.16 ± 3.57	32.50 ± 2.09	34.51 ± 0.66	2.55 ± 0.35	hs-CRP	Serum
Asker, 2015 [84]	Turkey	Caucasian	30/30	>18	34.35 ± 6.23	69.02 ± 29.04	>18	25.48 ± 2.29	2.23 ± 1.43	CRP	Serum
Bakırcı, 2015 [85]	Turkey	Caucasian	40/40	50.2 ± 7.6	29.2 ± 3.3	≥5	51.7 ± 8.3	28.6 ± 3.7	<5	hs-CRP	Serum
Kanbay, 2015 [81]	Turkey	Caucasian	64/19	53.91 ± 11.56	34.93 ± 5.58	44.41 ± 7.99	44.47 ± 13.37	31.6 ± 5.7	2.08 ± 1.3	hs-CRP	Serum
Korkmaz, 2015 [86]	Turkey	Caucasian	107/40	47 ± 9	32	≥5	43.30 ± 11.14	29.27	<5	CRP	Serum
Xu, 2015 [87]	China	Asian	137/78	58.33 ± 8.61	26.1 ± 2.17	≥5	57.35 ± 8.08	25.77 ± 1.29	<5	hs-CRP	Serum
Altintas, 2016 [88]	Turkey	Caucasian	40/40	54.86 ± 10.42	34.85 ± 6.22	53.43 ± 15.92	51.5 ± 6.7	32.9 ± 4.7	1.9 ± 1.4	CRP	Serum
Archontogeorgis, 2016 [89]	Greece	Caucasian	64/20	51.78 ± 11.55	36.34 ± 13.18	≥5	51.40 ± 16.24	33.73 ± 5.68	<5	CRP	Serum
Borratynska, 2016 [90]	Poland	Caucasian	110/55	57.33 ± 11.11	32.37 ± 6.89	22 ± 20	54.66 ± 10.37	28.95 ± 3.55	2 ± 2.96	hs-CRP	Plasma
Can, 2016 [91]	Turkey	Caucasian	23/27	56.2 ± 8.4	30.0 ± 3.8	34.0 ± 20.6	49.6 ± 11.7	28.8 ± 4.6	1.6 ± 1.2	CRP	Serum
Cao, 2016 [92]	China	Asian	192/56	53.41 ± 12.3	25.81 ± 3.97	21.80 ± 3.41	49.4 ± 11.6	24.2 ± 2.7	3.2 ± 1.3	CRP	Serum
Kim, 2016 [93]	Korea	Asian	862/973	57.45 ± 7.34	25.3 ± 2.8	13.41 ± 5.05	53.8 ± 6.6	23.9 ± 2.6	1.9 ± 1.4	hs-CRP	Plasma
Qi, 2016 [94]	China	Asian	96/10	52.0 ± 12.66	23.70 ± 1.50	31.38 ± 10.20	46.7 ± 8.68	23.71 ± 1.06	2.96 ± 2.31	hs-CRP	Serum
Tanrıverdi, 2016 [95]	Turkey	Caucasian	53/24	49.9 ± 8.8	31.6 ± 5.2	27.5 ± 22.7	44.2 ± 13.4	29.4 ± 4.6	1.73 ± 1.2	CRP	Serum
Tie, 2016 [96]	China	Asian	30/20	68.27 ± 8.32	26.61 ± 2.22	≥5	56.30 ± 8.52	25.73 ± 2.72	<5	CRP	Serum
Vicente, 2016 [97]	Spain	Caucasian	89/26	45.33 ± 14.81	30.03 ± 5.04	28 ± 23.70	45 ± 11.11	28.7 ± 4.37	1.9 ± 2.7	CRP	Plasma
Uygur, 2016 [98]	Turkey	Caucasian	96/31	51.4 ± 9.7	30.8 ± 3.7	27.9 ± 20.6	50.6 ± 12.8	29.6 ± 4.1	1.9 ± 1.7	CRP	Serum
Zhang, 2016 [99]	China	Asian	41/19	48.08 ± 7.14	24.77 ± 1.51	37.55 ± 4.62	47.45 ± 8.37	24.48 ± 1.66	3.65 ± 0.42	hs-CRP	Serum
Bouloukaki, 2017 [100]	Greece	Caucasian	858/190	43 ± 11.5	31 ± 8	44 ± 23	38.8 ± 14.1	26.6 ± 6	2 ± 2	hs-CRP	Serum
Gamsiz-Isik, 2017 [101]	Turkey	Caucasian	83/80	46.87 ± 8.21	31.53 ± 3.44	≥5	44.23 ± 9.83	30.91 ± 3.31	<5	hs-CRP	Serum
Karamanli, 2017 [102]	Turkey	Caucasian	68/30	47.2 ± 1.2	27.3 ± 3.4	34.7 ± 22.2	51.5 ± 1.3	26.2 ± 3.1	2.4 ± 1.5	CRP	Serum
Kosacka, 2017 [103]	Poland	Caucasian	163/59	55.41 ± 8.63	34.98 ± 7.55	35.02 ± 22.28	51.27 ± 12.97	29.47 ± 5.42	2.21 ± 1.90	CRP	Serum
Suliman, 2017 [104]	Egypt	Caucasian	43/17	50.2 ± 11.2	42.2 ± 6.5	≥5	46.8 ± 13.09	41.6 ± 3.3	<5	hs-CRP	Serum
Xu, 2017 [105]	China	Asian	33/30	51.6 ± 9.8	30.1 ± 3.5	19.6 ± 4.7	49.2 ± 13.1	28.9 ± 4.4	2.2 ± 1.5	hs-CRP	Serum
Bozic, 2018 [106]	Croatia	Caucasian	50/25	53.0 ± 11.9	28.9 ± 2.7	35.0 ± 11.0	52.5 ± 10.2	27.8 ± 2.2	<5	hs-CRP	Plasma
Bozkus, 2018 [107]	Turkey	Caucasian	167/39	47.75 ± 10.45	31.15 ± 5.82	33.77 ± 23.11	42.8 ± 10.02	24.50 ± 3.45	3.37 ± 1.15	CRP	Serum
Cengiz, 2018 [108]	Turkey	Caucasian	44/44	44 ± 10	31.27 ± 12.19	≥5	44 ± 12	32.23 ± 17.24	<5	CRP	Serum
Horvath, 2018 [109]	Hungary	Caucasian	50/26	61 ± 9	31 ± 6	49.1 ± 84.22	56 ± 8	26 ± 3	2.2 ± 3.55	CRP	Plasma
Kunos, 2018 [110]	Hungary	Caucasian	45/31	60 ± 11	31.0 ± 6.5	27.8 ± 21.6	53 ± 15	25.4 ± 3.6	2.3 ± 1.2	CRP	Serum
Ozkok, 2018 [111]	Turkey	Caucasian	120/31	52.48 ± 12.05	32.85 ± 5.7	40.09 ± 14.81	46 ± 13	30 ± 5	2.84 ± 1.41	hs-CRP	Serum
Ye, 2018 [112]	China	Asian	105/41	46 ± 9.5	28 ± 3.4	30.16 ± 12.80	46 ± 9	26.2 ± 3.1	2 ± 2	CRP	Serum
Zhang, 2018 [113]	China	Asian	30/20	40.73 ± 8.90	28.85 ± 2.62	61.48 ± 15.00	36.10 ± 13.67	27.55 ± 2.97	1.93 ± 1.38	hs-CRP	Plasma
Bhatt, 2019 [114]	India	Caucasian	47/25	44.2 ± 9.1	32.5 ± 6.9	13.5 ± 6.4	28.5 ± 8.6	41 ± 8.5	2.3 ± 1.1	CRP	Serum
Jung, 2019 [115]	Korea	Asian	87/21	45.76 ± 3.07	26.40 ± 2.87	≥5	47.1 ± 2.6	27.6 ± 8.1	<5	hs-CRP	Serum
Li, 2019 [116]	China	Asian	77/23	44.18 ± 12.18	26.82 ± 3.78	39.87 ± 25.66	43.78 ± 14.35	23.24 ± 3.43	2.47 ± 1.27	CRP	Serum
Płóciniczak, 2019 [117]	Poland	Asian	57/44	56.33 ± 11.11	31.57 ± 4.74	30.26 ± 30.96	50.66 ± 11.85	26.63 ± 4.29	2.03 ± 2.0	hs-CRP	Serum
Voulgaris, 2019 [118]	Greece	Caucasian	64/32	51 ± 12.2	35.9 ± 13.1	≥5	50.1 ± 11.7	33.9 ± 8.8	<5	CRP	Serum
Wang, 2019 [119]	China	Asian	72/58	53.6 ± 11.9	25.1 ± 2.9	16.43 ± 7.79	41.8 ± 14.5	24.2 ± 2.6	1.13 ± 1.11	hs-CRP	Serum
Wen, 2019 [120]	China	Asian	120/40	53.63 ± 11.8	26.63 ± 3.5	26.91 ± 9.38	46.9 ± 15.2	24.3 ± 3.7	2.8 ± 1.55	hs-CRP	Serum
Bocskei, 2020 [121]	Hungary	Caucasian	53/15	57.33 ± 11.11	32.37 ± 5.66	29.6 ± 17.92	47 ± 22.96	24.6 ± 4.58	1.66 ± 1.04	CRP	Plasma
Chen, 2020 [122]	China	Asian	73/17	42.68 ± 11.53	25.78 ± 2.71	52.1 ± 12.7	41.76 ± 11.71	25.54 ± 2.11	4.37 ± 2.18	hs-CRP	Serum
Chien, 2020 [123]	Taiwan	Asian	20/20	50.2 ± 5.6	26.05 ± 2.92	≥5	50.4 ± 6.7	25.82 ± 2.76	<5	hs-CRP	Serum
Düger, 2020 [124]	Turkey	Caucasian	86/83	45.1 ± 3.2	32.3 ± 5.9	≥5	42.8 ± 14	30.9 ± 2.3	<5	CRP	Serum
Pákó, 2020 [125]	UK	Caucasian	41/21	55.6 ± 13.2	27.5 ± 4.8	16.1 ± 10.1	48 ± 16	24.9 ± 4.7	1.9 ± 1.2	CRP	Serum
Winiarska, 2020 [126]	Poland	Caucasian	48/16	54.8 ± 10	30.60 ± 4.49	28.16 ± 5.27	49.83 ± 11.11	25.1 ± 3.85	1.86 ± 1.95	CRP	Serum
Xie, 2020 [127]	China	Asian	107/34	48.22 ± 13.0	27.84 ± 3.70	40.49 ± 12.69	34.74 ± 14.02	23.80 ± 4.00	2.23 ± 1.49	hs-CRP	Serum
Zhang, 2020 [128]	China	Asian	134/19	31 ± 7.7	42.95 ± 6.3	32.25 ± 13.25	27.8 ± 7.3	38.7 ± 3.5	2.8 ± 1.3	hs-CRP	Serum
**Children**											
Kaditis, 2010 [129]	Greece	Caucasian	84/22	6.05 ± 2.21	1.3 ± 1.23	6.37 ± 5.16	6.8 ± 2.6	− 0.1 ± 1.5	0.6 ± 0.2	CRP	Plasma
Kheirandish-Gozal, 2010 [130]	USA	Mixed	80/20	7.2 ± 1.4	0.96 ± 0.3	12.9 ± 8.5	7.1 ± 1.6	0.56 ± 0.2	0.4 ± 0.3	hs-CRP	Serum
Kim, 2010 [4]	USA	Mixed	140/115	7.54 ± 1.58	1.47 ± 1.31	5.71 ± 3.41	7.81 ± 1.44	1.15 ± 1.22	0.40 ± 0.27	hs-CRP	Plasma
Canapari, 2011 [131]	USA	Mixed	15/16	12.7 ± 2.64	2.78 ± 0.39	6.26 ± 6.77	12.6 ± 2.73	2.44 ± 0.27	0.48 ± 0.30	CRP	Serum
Khalyfa, 2012 [132]	USA	Mixed	131/323	7.03 ± 0.1	1.11 ± 1.5	8.13 ± 2.4	7.14 ± 0.1	0.78 ± 1.2	0.32 ± 0.0	hs-CRP	Plasma
Kim, 2013 [133]	USA	Mixed	62/44	8.13 ± 1.75	1.61 ± 1.17	≥1	8.4 ± 1.4	1.35 ± 1.01	<1	hs-CRP	Plasma
Iannuzzi, 2013 [134]	Italy	Caucasian	19/25	9.51 ± 2.35	25.5 ± 7.0	≥1	10.65 ± 2.11	23.6 ± 7.8	<1	hs-CRP	Plasma
Israel, 2013 [135]	Israel	Mixed	25/24	5.1 ± 3.2	0.62 ± 1.04	14.1 ± 2.9	5.3 ± 3.5	0.57 ± 1.11	0.6 ± 0.2	hs-CRP	Serum
Gozal, 2014 [136]	USA	Mixed	138/88	6.85 ± 2.0	1.21 ± 0.18	≥1	7.25 ± 2.05	1.19 ± 0.71	<1	hs-CRP	Plasma
Kheirandish-Gozal, 2014 [137]	USA	Mixed	110/109	6.85 ± 1.4	1.21 ± 0.18	9.0 ± 10	6.85 ± 1.55	1.19 ± 0.71	0.4 ± 0.4	hs-CRP	Plasma
Ye, 2015 [138]	China	Asian	25/19	6.45 ± 2.84	1.28 ± 0.64	34.76 ± 15.28	6.63 ± 2.71	1.25 ± 0.47	0.38 ± 0.20	hs-CRP	Serum
Huang, 2016 [139]	Taiwan	Asian	47/32	7.84 ± 0.56	0.15 ± 0.21	9.13 ± 1.67	7.02 ± 0.65	−0.12 ± 0.27	0.37 ± 0.06	hs-CRP	Serum
Smith, 2017 [3]	USA	Mixed	65/90	9.2 ± 2.6	1.1 ± 1.25	11.06 ± 7.99	9.7 ± 2.5	0.7 ± 1	0.4 ± 0.3	CRP	Serum

Abbreviations: NR, Not reported; OSA, Obstructive sleep apnea; AHI, Apnea-hypopnea index; BMI, Body mass index (based on kg/m^2^ in adults and Z-score in children); CRP, C-reactive protein; hs-CRP, High-sensitivity CRP.

**Table 2 life-11-00362-t002:** The results of forest plot analysis of plasma high-sensitivity C-reactive protein (hs-CRP) levels in adults.

Variable	Studies	OSA			Control			Weight	Mean Difference IV, Random, 95% CI
		Mean	SD	Total	Mean	SD	Total		
Plasma hs-CRP	Shamsuzzaman, 2002 [33]	0.87	0.66	20	0.28	0.22	20	2.2%	0.59 [0.29, 0.89]
	Teramoto, 2003 [34]	0.31	0.13	40	0.12	0.06	40	11.5%	0.19 [0.15, 0.23]
	Chung, 2007 [43]	0.12	0.14	68	0.063	0.083	22	11.3%	0.06 [0.01, 0.11]
	Takahashi, 2008 [50]	0.172	0.141	41	0.087	0.096	12	10.1%	0.08 [0.02, 0.15]
	Carneiro, 2009 [52]	0.83	0.14	16	0.91	0.34	13	4.2%	−0.08 [−0.28, 0.12]
	Sahlman, 2010 [55]	0.167	0.253	84	0.13	0.253	40	8.6%	0.04 [−0.06, 0.13]
	Chen, 2013 [69]	0.057	0.069	44	0.01	0.0148	20	12.4%	0.05 [0.03, 0.07]
	Borratynska, 2016 [90]	0.243	0.207	110	0.157	0.17	55	10.7%	0.09 [0.03, 0.15]
	Kim, 2016 [93]	1.28	1.41	862	0.97	1.22	973	7.2%	0.31 [0.19, 0.43]
	Bozic, 2018 [106]	0.287	0.067	50	0.129	0.031	25	12.4%	0.16 [0.14, 0.18]
	Zhang, 2018 [113]	0.209	0.18	30	0.119	0.114	20	9.4%	0.09 [0.01, 0.17]
Total (95% CI)				1365			1240	100.0%	0.11 [0.07, 0.16]
Heterogeneity: Tau^2^ = 0.00; Chi^2^ = 94.93, df = 10 (*p* < 0.00001); I^2^ = 89%; Test for overall effect: Z = 4.57 (*p* < 0.00001)									

Abbreviations: SD, standard deviation; OSA, obstructive sleep apnea, CI, confidence interval.

**Table 3 life-11-00362-t003:** The results of forest plot analysis of serum high-sensitivity C-reactive protein (hs-CRP) levels in adults.

Variable	Studies	OSA			Control			Weight	Mean Difference IV, Random, 95% CI
		Mean	SD	Total	Mean	SD	Total		
Serum hs-CRP	Yokoe, 2003 [35]	0.21	0.1	26	0.07	0.037	14	3.5%	0.14 [0.10, 0.18]
	Minoguchi, 2006 [40]	0.16	0.15	40	0.06	0.078	30	3.2%	0.10 [0.05, 0.15]
	Iesato, 2007 [44]	0.152	0.011	155	0.072	0.011	39	4.2%	0.08 [0.08, 0.08]
	Ryan, 2007 [42]	0.228	0.183	66	0.132	0.1	30	3.1%	0.10 [0.04, 0.15]
	Minoguchi, 2007 [45]	0.21	0.19	50	0.1	0.08	15	2.8%	0.11 [0.04, 0.18]
	Sharma, 2008 [49]	0.51	0.37	29	0.46	0.44	68	1.0%	0.05 [−0.12, 0.22]
	Saletu, 2008 [48]	0.478	0.621	103	0.28	0.46	44	0.9%	0.20 [0.02, 0.38]
	Bhushan, 2009 [51]	0.36	0.2	62	0.14	0.14	46	2.9%	0.22 [0.16, 0.28]
	Cofta, 2009 [53]	0.231	0.116	40	0.2	0.102	14	2.9%	0.03 [−0.03, 0.10]
	Aihara, 2011 [56]	0.181	0.298	150	0.14	0.27	20	1.5%	0.04 [−0.09, 0.17]
	Kasai, 2011 [62]	0.192	0.177	50	0.129	0.117	25	2.8%	0.06 [−0.00, 0.13]
	Guasti, 2011 [60]	0.298	0.27	16	0.481	0.472	11	0.4%	−0.18 [−0.49, 0.13]
	Basoglu, 2011 [58]	0.4	0.2	36	0.3	0.1	34	2.6%	0.10 [0.03, 0.17]
	Fredheim, 2011 [59]	2.6	2.1	84	3.5	3.2	53	0.0%	−0.90 [−1.87, 0.07]
	Chien, 2012 [64]	0.207	0.092	30	0.102	0.057	30	3.6%	0.10 [0.07, 0.14]
	Balci, 2012 [63]	4.25	2.45	61	1.6	0.7	33	0.1%	2.65 [1.99, 3.31]
	Guven, 2012 [67]	0.403	0.358	47	0.241	0.195	29	1.5%	0.16 [0.04, 0.29]
	Zhang, 2013 [72]	0.0997	0.0268	75	0.088	0.02	23	4.2%	0.01 [0.00, 0.02]
	Ciccone, 2014 [74]	0.167	0.061	80	0.108	0.053	40	4.0%	0.06 [0.04, 0.08]
	Yadav, 2014 [79]	0.75	0.474	20	0.76	0.259	21	0.6%	−0.01 [−0.25, 0.23]
	Yüksel, 2014 [80]	6.0	3.6	44	1.0	0.7	49	0.0%	5.00 [3.92, 6.08]
	Akilli, 2014 [73]	3.18	2.56	149	3.0	2.54	50	0.1%	0.18 [−0.64, 1.00]
	Shi, 2014 [77]	0.943	0.525	126	0.593	0.333	74	1.6%	0.35 [0.23, 0.47]
	Korkmaz, 2015 [86]	0.59	1.01	107	0.31	0.18	40	0.8%	0.28 [0.08, 0.48]
	da Silva Araújo, 2015 [83]	0.055	0.0090	33	0.046	0.0070	20	4.2%	0.01 [0.00, 0.01]
	Kanbay, 2015 [81]	1.023	0.598	64	0.505	0.296	19	0.8%	0.52 [0.32, 0.72]
	Andaku, 2015 [82]	0.21	0.06	11	0.11	0.08	10	3.0%	0.10 [0.04, 0.16]
	Xu, 2015 [87]	0.107	0.081	137	0.055	0.034	78	4.1%	0.05 [0.04, 0.07]
	Bakırcı, 2015 [85]	0.13	0.05	40	0.1	0.03	40	4.1%	0.03 [0.01, 0.05]
	Zhang, 2016 [99]	0.425	0.061	41	0.332	0.035	19	4.0%	0.09 [0.07, 0.12]
	Qi, 2016 [94]	0.113	0.112	96	0.157	0.234	10	1.2%	−0.04 [−0.19, 0.10]
	Gamsiz-Isik, 2017 [101]	0.495	0.895	83	0.238	0.18	80	0.8%	0.26 [0.06, 0.45]
	Suliman, 2017 [104]	3.41	4.52	43	0.6	0.89	17	0.0%	2.81 [1.39, 4.23]
	Xu, 2017 [105]	0.147	0.16	33	0.097	0.122	30	2.7%	0.05 [−0.02, 0.12]
	Bouloukaki, 2017 [100]	0.539	1.07	858	0.367	0.592	190	1.8%	0.17 [0.06, 0.28]
	Ozkok, 2018 [111]	0.371	0.501	120	0.143	0.281	31	1.4%	0.23 [0.09, 0.36]
	Wang, 2019 [119]	0.209	0.246	72	0.14	0.059	58	3.1%	0.07 [0.01, 0.13]
	Jung, 2019 [115]	0.048	0.057	87	0.038	0.033	21	4.1%	0.01 [−0.01, 0.03]
	Chen, 2020 [122]	0.113	0.03	73	0.081	0.022	17	4.2%	0.03 [0.02, 0.04]
	Wen, 2019 [120]	0.183	0.281	120	0.107	0.133	40	2.9%	0.08 [0.01, 0.14]
	Płóciniczak, 2019 [117]	0.203	0.148	57	0.133	0.111	44	3.3%	0.07 [0.02, 0.12]
	Chien, 2020 [123]	0.23	0.14	20	0.12	0.06	20	2.8%	0.11 [0.04, 0.18]
	Zhang, 2020 [128]	1.0	0.693	134	0.8	0.54	19	0.5%	0.20 [−0.07, 0.47]
	Xie, 2020 [127]	0.282	0.331	107	0.082	0.121	34	2.6%	0.20 [0.13, 0.27]
Total (95% CI)				3875			1629	100.0%	0.09 [0.07, 0.11]
Heterogeneity: Tau^2^ = 0.00; Chi^2^ = 989.97, df = 43 (*p* < 0.00001); I^2^ = 96%; Test for overall effect: Z = 9.49 (*p* < 0.00001)									

Abbreviations: SD, standard deviation; OSA, obstructive sleep apnea, CI, confidence interval.

**Table 4 life-11-00362-t004:** The results of forest plot analysis of plasma C-reactive protein (CRP) levels in adults.

Variable	Studies	OSA			Control			Weight	Mean Difference IV, Random, 95% CI
		Mean	SD	Total	Mean	SD	Total		
Plasma CRP	Barceló, 2004 [36]	19.27	3.05	47	7.24	2.81	18	2.9%	12.03 [10.47, 13.59]
	Shiina, 2006 [41]	1.5	0.94	94	1.1	0.97	90	11.6%	0.40 [0.12, 0.68]
	Makino, 2009 [54]	0.118	0.136	157	0.039	0.039	24	12.8%	0.08 [0.05, 0.11]
	Barceló, 2011 [57]	2.17	2.0	119	7.0	1.98	119	9.4%	−4.83 [−5.34,−4.32]
	Panoutsopoulos, 2012 [68]	0.82	0.16	20	0.29	0.14	18	12.7%	0.53 [0.43, 0.63]
	Niżankowska-Jędrzejczyk, 2014 [76]	0.127	0.141	22	0.108	0.069	16	12.8%	0.02 [−0.05, 0.09]
	Vicente, 2016 [97]	0.533	0.211	30	0.235	0.09	20	12.7%	0.30 [0.21, 0.38]
	Horvath, 2018 [109]	0.42	0.37	50	0.4	0.18	26	12.6%	0.02 [−0.10, 0.14]
	Bocskei, 2020 [121]	0.234	0.243	53	0.24	0.224	15	12.5%	−0.01 [−0.14, 0.12]
Total (95% CI)				592			346	100.0%	0.06 [−0.24, 0.36]
Heterogeneity: Tau^2^ = 0.18; Chi^2^ = 704.06, df = 8 (*p* < 0.00001); I^2^ = 99%; Test for overall effect: Z = 0.36 (*p* = 0.72)									

Abbreviations: SD, standard deviation; OSA, obstructive sleep apnea, CI, confidence interval.

**Table 5 life-11-00362-t005:** The results of forest plot analysis of serum C-reactive protein (CRP) levels in adults.

Variable	Studies	OSA			Control			Weight	Mean Difference IV, Random, 95% CI
		Mean	SD	Total	Mean	SD	Total		
Serum CRP	Minoguchi, 2005	2.1	1.95	36	0.09	0.02	16	1.3%	2.01 [1.37, 2.65]
	Can, 2006	3.9	1.93	62	1.8	0.61	30	1.7%	2.10 [1.57, 2.63]
	Jin, 2007	3.68	0.94	51	1.4	0.9	25	2.1%	2.28 [1.84, 2.72]
	Kapsimalis, 2008	0.31	0.25	52	0.19	0.1	15	4.3%	0.12 [0.04, 0.20]
	Kanbay, 2011	8.11	4.72	144	4.8	1.7	22	0.6%	3.31 [2.26, 4.36]
	Feng, 2012	0.276	0.091	132	0.384	0.125	108	4.5%	−0.11 [−0.14, −0.08]
	Fornadi, 2012	3.73	3.11	25	3.41	3.41	75	0.3%	0.32 [−1.12, 1.76]
	Wang, 2013	0.36	0.214	192	0.224	0.145	144	4.5%	0.14 [0.10, 0.17]
	Kosacka, 2013	6.58	6.52	137	4.09	2.79	42	0.4%	2.49 [1.11, 3.87]
	Sökücü, 2014	0.498	0.404	36	0.117	0.185	22	4.0%	0.38 [0.23, 0.53]
	Li, 2014	0.361	0.176	156	0.226	0.15	110	4.5%	0.13 [0.10, 0.17]
	Asker, 2015	0.0919	0.0872	30	0.049	0.052	30	4.5%	0.04 [0.01, 0.08]
	Guilleminault, 2004	0.464	0.674	146	0.41	0.21	54	4.1%	0.05 [−0.07, 0.18]
	Tanrıverdi, 2016	0.33	0.281	53	0.271	0.25	24	4.1%	0.06 [−0.07, 0.18]
	Tie, 2016	0.533	0.211	30	0.235	0.09	20	4.3%	0.30 [0.21, 0.38]
	Uygur, 2016	0.36	0.18	96	0.14	0.09	20	4.5%	0.22 [0.17, 0.27]
	Altintas, 2016	3.63	5.63	40	2.8	3.26	40	0.2%	0.83 [−1.19, 2.85]
	Cao, 2016	1.238	0.271	192	0.92	0.12	56	4.5%	0.32 [0.27, 0.37]
	Can, 2016	0.536	0.308	23	0.26	0.21	27	4.0%	0.28 [0.13, 0.42]
	Archontogeorgis, 2016	0.55	0.57	64	0.32	0.41	20	3.4%	0.23 [0.00, 0.46]
	Kosacka, 2017	0.655	0.624	163	0.27	0.15	30	4.2%	0.39 [0.28, 0.49]
	Karamanli, 2017	0.76	0.13	68	0.27	0.15	30	4.4%	0.49 [0.43, 0.55]
	Ye, 2018	1.8	0.461	105	1.46	0.41	41	4.0%	0.34 [0.19, 0.49]
	Cengiz, 2018	0.21	0.477	44	0.155	1.32	44	2.2%	0.05 [−0.36, 0.47]
	Kunos, 2018	0.63	1.3	45	0.28	0.24	31	2.3%	0.35 [−0.04, 0.74]
	Bozkus, 2018	3.72	1.36	167	3.12	0.62	39	3.0%	0.60 [0.32, 0.88]
	Bhatt, 2019	3.6	1.5	47	1.4	0.7	25	1.7%	2.20 [1.69, 2.71]
	Voulgaris, 2019	0.55	0.58	64	0.34	0.36	32	3.7%	0.21 [0.02, 0.40]
	Li, 2019	0.321	0.239	77	0.252	0.431	23	3.7%	0.07 [−0.12, 0.25]
	Düger, 2020	0.277	0.244	86	0.187	0.192	83	4.4%	0.09 [0.02, 0.16]
	Winiarska, 2020	0.165	0.122	48	0.093	0.059	16	4.5%	0.07 [0.03, 0.12]
	Pákó, 2020	10.3	22.7	41	4.5	12.1	21	0.0%	5.80 [−2.86, 14.46]
Total (95% CI)				2652			1315	100.0%	0.36 [0.28, 0.45]
Heterogeneity: Tau^2^ = 0.04; Chi^2^ = 859.49, df = 31 (*p* < 0.00001); I^2^ = 96%; Test for overall effect: Z = 8.36 (*p* < 0.00001)									

Abbreviations: SD, standard deviation; OSA, obstructive sleep apnea, CI, confidence interval.

**Table 6 life-11-00362-t006:** Forest plot of random- or fixed-effects analysis of plasma and serum levels of high-sensitivity C-reactive protein (hs-CRP) and CRP in children.

Variable	Studies	OSA			Control			Weight	Mean Difference IV, Random, 95% CI
		Mean	SD	Total	Mean	SD	Total		
Plasma hs-CRP	Kim, 2010 [4]	1.75	2.26	140	1.16	1.57	115	17.1%	0.59 [0.12, 1.06]
	Khalyfa, 2012 [132]	2.7	4.2	131	1.8	3.4	323	15.4%	0.90 [0.09, 1.71]
	Iannuzzi, 2013 [134]	0.258	0.367	19	0.098	0.495	25	17.9%	0.16 [−0.09, 0.41]
	Kim, 2013 [133]	1.54	1.8	62	0.96	1.27	44	16.6%	0.58 [−0.00, 1.16]
	Kheirandish-Gozal, 2014 [137]	4.13	3.82	110	0.775	0.638	109	15.9%	3.35 [2.63, 4.08]
	Gozal, 2014 [136]	3.1	2.1	138	1.53	1.59	88	17.1%	1.57 [1.09, 2.05]
Total (95% CI)				600			704	100.0%	1.17 [0.35, 1.98]
Heterogeneity: Tau^2^ = 0.95; Chi^2^ = 82.02, df = 5 (*p* < 0.00001); I^2^ = 94%; Test for overall effect: Z = 2.80 (*p* = 0.005)									
Serum hs-CRP	Huang, 2016 [139]	0.19	0.044	47	0.041	0.48	32	21.7%	0.15 [−0.02, 0.32]
	Israel, 2013 [135]	0.45	0.21	25	0.15	0.1	24	25.1%	0.30 [0.21, 0.39]
	Kheirandish-Gozal, 2010 [137]	0.29	0.17	80	0.04	0.07	20	26.4%	0.25 [0.20, 0.30]
	Ye, 2015 [138]	0.011	0.0021	25	0.0023	9.0 × 10^−4^	19	26.9%	0.01 [0.01, 0.01]
Total (95% CI)				177			95	100.0%	0.18 [−0.00, 0.35]
Heterogeneity: Tau^2^ = 0.03; Chi^2^ = 137.61, df = 3 (*p* < 0.00001); I^2^ = 98%; Test for overall effect: Z = 1.96 (*p* = 0.05)									
Plasma hs-CRP	Kaditis, 2010 [129]	0.213	0.336	84	0.13	0.16	22	100.0%	0.08 [−0.02, 0.18]
Total (95% CI)				84			22	100.0%	0.08 [−0.02, 0.18]
Heterogeneity: Not applicable; Test for overall effect: Z = 1.66 (*p* = 0.10)									
Serum hs-CRP	Canapari, 2011 [131]	0.557	0.558	15	0.382	0.195	16	7.9%	0.18 [−0.12, 0.47]
	Smith, 2017 [3]	0.11	0.2	78	0.08	0.28	53	92.1%	0.03 [−0.06, 0.12]
Total (95% CI)				93			69	100.0%	0.04 [−0.04, 0.13]
Heterogeneity: Chi^2^ = 0.84, df = 1 (*p* = 0.36); I^2^ = 0%; Test for overall effect: Z = 0.97 (*p* = 0.33)									

Abbreviations: OSA, Obstructive sleep apnea; CI, Confidence interval; C-reactive protein; hs-CRP, High-sensitivity CRP; SD, Standard deviation. All analyses were performed based on random-effects model, except serum hs-CRP was based on fixed-effects model.

**Table 7 life-11-00362-t007:** Subgroup analysis on serum and plasma levels of high-sensitivity C-reactive protein (hs-CRP) in adult participants.

Subgroup Analysis of Plasma Level (N)	MD (95% CI), *p*-Value, I^2^ (%), *p*_h_	Subgroup Analysis of Serum Level (N)	MD (95% CI), *p*-Value, I^2^ (%), *p*_h_
Overall (11)		Overall (44)	
Ethnicity		Ethnicity	
Caucasian (3)	**0.10 (0.03, 0.18), 0.004, 80 (0.007)**	Caucasian (21)	**0.18 (0.11, 0.23), <0.00001, 92 (<0.00001)**
Asian (6)	**0.12 (0.05, 0.18), 0.0003, 89 (<0.00001)**	Asian (21)	**0.08 (0.06, 0.10), <0.00001, 93 (<0.00001)**
Mixed (2)	0.24 (−0.41, 0.90), 0.47, 92 (0.0003)	Mixed (2)	0.05 (−0.04, 0.14), 0.28, 88 (0.004)
Mean BMI of OSA patients, kg/m^2^		Mean BMI of OSA patients, kg/m^2^	
>30 (4)	0.10 (−0.04, 0.24), 0.15, 79, (0.003)	>30 (19)	**0.18 (0.09, 0.27), <0.0001, 92 (<0.00001)**
≤30 (6)	**0.11 (0.05, 0.17), 0.0003, 92 (<0.00001)**	≤30 (25)	**0.08 (0.06, 0.10), <0.00001, 93 (<0.00001)**
Mean BMI of controls, kg/m^2^		Mean BMI of controls, kg/m^2^	
>30 (3)	0.15 (−0.13, 0.43), 0.30, 85 (0.001)	>30 (12)	**0.11 (0.04, 0.19), 0.004, 85 (<0.00001)**
≤30 (7)	**0.11 (0.05, 0.16), <0.0001, 91 (<0.00001)**	≤30 (32)	**0.09 (0.07, 0.11), <0.00001, 94 (<0.00001)**
Total number of participants		Total number of participants	
>100 (3)	**0.14 (0.00, 0.27), 0.04, 85 (0.001)**	>100 (20)	**0.10 (0.08, 0.13), <0.00001, 86 (<0.00001)**
≤100 (8)	**0.11 (0.05, 0.17), 0.0002, 91 (<0.00001)**	≤100 (24)	**0.08 (0.06, 0.11), <0.00001, 93 (<0.00001)**
Mean AHI of OSA patients, events/h		Mean AHI of OSA patients, events/h	
>30 (4)	**0.14 (0.02, 0.26), 0.02, 81 (0.001)**	>30 (21)	**0.11 (0.08, 0.14), <0.00001, 92 (<0.00001)**
≤30 (4)	**0.10 (0.02, 0.18), 0.01, 84 (0.0003)**	≤30 (23)	**0.07 (0.05, 0.09), <0.00001, 88 (<0.00001)**

Abbreviations: BMI, Body mass index; AHI, Apnea–hypopnea index; CI, Confidence interval; OSA, Obstructive sleep apnea; OR, Odds ratio; N, number of studies; *p*_h_, *p*_heterogeneity._ Bold numbers show statistically significant value (*p*-value < 0.05).

**Table 8 life-11-00362-t008:** Subgroup analysis on serum and plasma levels of C-reactive protein (CRP) in adult participants.

Subgroup Analysis of Plasma Level (N)	MD (95% CI), *p*-Value, I^2^ (%), *p*_h_	Subgroup Analysis of Serum Level (N)	MD (95% CI), *p*-Value, I^2^ (%), *p*_h_
Overall (9)	0.06 (−0.24, 0.36), 0.72, 99 (<0.00001)	Overall (32)	**0.36 (0.28, 0.45), <0.00001, 96 (<0.00001)**
Ethnicity		Ethnicity	
Caucasian (7)	0.22 (−0.28, 0.71), 0.39, 99 (<0.00001)	Caucasian (21)	**0.38 (0.27, 0.50), <0.00001, 95 (<0.00001)**
Asian (2)	0.21 (−0.10, 0.52), 0.18, 81 (0.02)	Asian (9)	**0.38 (0.23, 0.54), <0.00001, 98 (<0.00001)**
Mixed (0)	-	Mixed (2)	0.06 (−0.07, 0.18), 0.37, 0 (0.72)
Mean BMI of OSA patients, kg/m^2^		Mean BMI of OSA patients, kg/m^2^	
>30 (5)	**0.95 (0.47, 1.44), 0.0001, 99 (<0.00001)**	>30 (16)	**0.31 (0.21, 0.42), <0.00001, 92 (<0.00001)**
≤30 (4)	**−0.88 (−1.48, −0.27), 0.004, 99 (<0.00001)**	≤30 (16)	**0.41 (0.28, 0.55), <0.00001, 98 (<0.00001)**
Mean BMI of controls, kg/m^2^		Mean BMI of controls, kg/m^2^	
>30 (0)	-	>30 (7)	**0.61 (0.23, 1.00), 0.002, 92 (<0.00001)**
≤30 (9)	0.06 (−0.24, 0.36), 0.72, 99 (<0.00001)	≤30 (25)	**0.35 (0.26, 0.44), <0.00001, 97 (<0.00001)**
Total number of participants		Total number of participants	
>100 (3)	−1.43 (−3.34, 0.48), 0.14, 99 (<0.00001)	>100 (12)	**0.21 (0.09, 0.32), 0.0003, 97 (<0.000001)**
≤100 (6)	**0.69 (0.32, 1.05), 0.0003, 98 (<0.00001)**	≤100 (20)	**0.53 (0.39, 0.67), <0.00001, 96 (<0.00001)**
Mean AHI of OSA patients, events/h		Mean AHI of OSA patients, events/h	
>30 (4)	1.64 (−0.63, 3.92), 0.16, 99 (<0.00001)	>30 (13)	**0.54 (0.36, 0.72), <0.00001, 96 (<0.00001)**
≤30 (5)	**0.18 (0.02, 0.34), 0.02, 96 (<0.00001)**	≤30 (19)	**0.27 (0.17, 0.37), <0.00001, 96 (<0.00001)**

Abbreviations: BMI, Body mass index; AHI, Apnea–hypopnea index; CI, Confidence interval; OSA, Obstructive sleep apnea; OR, Odds ratio; N, number of studies; *p*_h_, *p*_heterogeneity._ Bold numbers show statistically significant value (*p*-value < 0.05).

**Table 9 life-11-00362-t009:** Meta-regression analysis based on some variables for serum and plasma levels of high-sensitivity C-reactive protein (hs-CRP) in obstructive sleep apnea patients compared with controls in adult participants.

Year of Publication	R	Adjusted R^2^	*p*	Mean Age of OSA Patients	R	Adjusted R^2^	*p*	Mean Age of Controls	R	Adjusted R^2^	*p*
Plasma	0.315	−0.001	0.346	Plasma	0.357	0.019	0.311	Plasma	0.226	−0.068	0.531
Serum	0.062	−0.020	0.691	Serum	0.041	−0.022	0.790	Serum	0.002	−0.024	0.989
**Mean BMI of OSA Patients**	**R**	**Adjusted R^2^**	***p***	**Mean BMI of Controls**	**R**	**Adjusted R^2^**	***p***	**Mean AHI of OSA Patients**	**R**	**Adjusted R^2^**	***p***
Plasma	0.147	−0.101	0.686	Plasma	0.222	−0.069	0.537	Plasma	0.130	−0.147	0.759
Serum	0.054	−0.021	0.726	Serum	0.010	−0.024	0.946	Serum	0.433	0.160	**0.013**
**Number of Participants**	**R**	**Adjusted R^2^**	***p***								
Plasma	0.298	−0.013	0.374								
Serum	0.076	−0.018	0.626								

Abbreviations: BMI, Body mass index; AHI, Apnea–hypopnea index; OSA, Obstructive sleep apnea. R, correlation coefficient. Bold numbers for statistically significant values (p < 0.05).

**Table 10 life-11-00362-t010:** Meta-regression analysis based on some variables for serum and plasma levels of C-reactive protein (CRP) in obstructive sleep apnea patients compared with controls in adult participants.

Year of Publication	R	Adjusted R^2^	*p*	Mean Age of OSA Patients	R	Adjusted R^2^	*p*	Mean Age of Controls	R	Adjusted R^2^	*p*
Plasma	0.504	0.147	0.166	Plasma	0.039	−0.141	0.920	Plasma	0.113	−0.128	0.772
Serum	0.187	0.003	0.306	Serum	0.116	−0.020	0.533	Serum	0.066	−0.030	0.724
**Mean BMI of OSA Patients**	**R**	**Adjusted R^2^**	***p***	**Mean BMI of Controls**	**R**	**Adjusted R^2^**	***p***	**Mean AHI of OSA Patients**	**R**	**Adjusted R^2^**	***p***
Plasma	0.235	−0.080	0.543	Plasma	0.294	−0.044	0.443	Plasma	0.316	−0.029	0.408
Serum	0.042	−0.032	0.820	Serum	0.058	−0.030	0.754	Serum	0.332	0.068	0.121
**Number of Participants**	**R**	**Adjusted R^2^**	***p***								
Plasma	0.403	0.043	0.282								
Serum	0.178	−0.001	0.331								

Abbreviations: BMI, Body mass index; AHI, Apnea–hypopnea index; OSA, Obstructive sleep apnea. R, correlation coefficient. Bold numbers for statistically significant values (p < 0.05).

**Table 11 life-11-00362-t011:** Quality assessment scores of the studies included in the meta-analysis.

The First Author, Year	Selection	Comparability	Exposure	Total Points
Adults				
Shamsuzzaman, 2002 [33]	***	**	***	8
Teramoto, 2003 [34]	**	*	***	6
Yokoe, 2003 [35]	***	**	***	8
Barceló, 2004 [36]	***	**	***	8
Guilleminault, 2004 [37]	****	*	***	8
Minoguchi, 2005 [38]	***	**	***	8
Can, 2006 [39]	***	**	***	8
Minoguchi, 2006 [40]	***	**	***	8
Shiina, 2006 [41]	****	**	***	9
Ryan, 2007 [42]	***	**	***	8
Chung, 2007 [43]	***	**	***	8
Iesato, 2007 [44]	****	**	***	9
Minoguchi, 2007 [45]	****	**	***	9
Jin, 2007 [46]	***	**	***	8
Kapsimalis, 2008 [47]	****	**	***	9
Saletu, 2008 [48]	***	**	***	8
Sharma, 2008 [49]	****	**	***	9
Takahashi, 2008 [50]	***	**	***	8
Bhushan, 2009 [51]	***	**	***	8
Carneiro, 2009 [52]	***	**	***	8
Cofta, 2009 [53]	***	**	***	8
Makino, 2009 [54]	***	**	***	8
Sahlman, 2010 [55]	***	**	***	8
Aihara, 2011 [56]	***	*	***	8
Barceló, 2011 [57]	***	**	***	8
Basoglu, 2011 [58]	***	**	***	8
Fredheim, 2011 [59]	***	*	***	8
Guasti, 2011 [60]	****	**	***	9
Kanbay, 2011 [61]	***	**	***	8
Kasai, 2011 [62]	***	*	***	7
Balci, 2012 [63]	***	**	***	8
Chien, 2012 [64]	***	**	***	8
Feng, 2012 [65]	***	**	***	8
Fornadi, 2012 [66]	***	**	***	8
Guven, 2012 [67]	***	**	***	8
Panoutsopoulos, 2012 [68]	***	**	***	8
Chen, 2013 [69]	***	*	***	7
Kosacka, 2013 [70]	***	**	***	8
Wang, 2013 [71]	***	**	***	8
Zhang, 2013 [72]	***	**	***	8
Akilli, 2014 [73]	***	**	***	8
Ciccone, 2014 [74]	***	**	***	8
Li, 2014 [75]	***	**	***	8
Niżankowska-Jędrzejczyk, 2014 [76]	***	**	***	8
Shi, 2014 [77]	***	**	***	8
Sökücü, 2014 [78]	***	*	***	7
Yadav, 2014 [79]	****	**	***	9
Yüksel, 2014 [80]	***	*	***	7
Abakay, 2015 [81]	***	**	***	8
Andaku, 2015 [82]	***	**	***	8
da Silva Araújo, 2015 [83]	***	**	***	8
Asker, 2015 [84]	***	-	***	6
Bakırcı, 2015 [85]	***	**	***	8
Kanbay, 2015 [81]	***	**	***	8
Korkmaz, 2015 [86]	***	**	***	8
Xu, 2015 [87]	***	**	***	8
Altintas, 2016 [88]	***	**	***	8
Archontogeorgis, 2016 [89]	***	**	***	8
Borratynska, 2016 [90]	****	**	***	9
Can, 2016 [91]	***	**	***	8
Cao, 2016 [92]	***	**	***	8
Kim, 2016 [93]	***	**	***	8
Qi, 2016 [94]	***	**	***	8
Tanrıverdi, 2016 [95]	***	**	***	8
Tie, 2016 [96]	***	**	***	8
Vicente, 2016 [97]	***	**	***	8
Uygur, 2016 [98]	***	**	***	8
Zhang, 2016 [99]	***	**	***	8
Bouloukaki, 2017 [100]	***	*	***	7
Gamsiz-Isik, 2017 [101]	***	**	***	8
Karamanli, 2017 [102]	***	**	***	8
Kosacka, 2017 [103]	***	**	***	8
Suliman, 2017 [104]	***	**	***	8
Xu, 2017 [105]	****	**	***	9
Bozic, 2018 [106]	***	**	***	8
Bozkus, 2018 [107]	***	*	***	7
Cengiz, 2018 [108]	***	**	***	8
Horvath, 2018 [109]	***	*	***	7
Kunos, 2018 [110]	***	*	***	7
Ozkok, 2018 [111]	***	**	***	8
Ye, 2018 [112]	***	**	***	8
Zhang, 2018 [113]	***	**	***	8
Bhatt, 2019 [114]	***	-	***	6
Jung, 2019 [115]	***	**	***	8
Li, 2019 [116]	***	**	***	8
Płóciniczak, 2019 [117]	***	*	***	7
Voulgaris, 2019 [118]	***	**	***	8
Wang, 2019 [119]	***	**	***	8
Wen, 2019 [120]	***	**	***	8
Bocskei, 2020 [121]	***	*	***	7
Chen, 2020 [122]	***	**	***	8
Chien, 2020 [123]	***	**	***	8
Düger, 2020 [124]	***	**	***	8
Pákó, 2020 [125]	***	**	***	8
Winiarska, 2020 [126]	***	**	***	8
Xie, 2020 [127]	***	*	***	8
Zhang, 2020 [128]	****	**	***	9
**Children**				
Kaditis, 2010 [129]	***	*	***	7
Kheirandish-Gozal, 2010 [130]	***	*	***	7
Kim, 2010 [4]	****	*	***	8
Canapari, 2011 [131]	***	*	***	7
Khalyfa, 2012 [132]	****	**	***	9
Kim, 2013 [133]	****	**	***	9
Iannuzzi, 2013 [134]	****	**	***	9
Israel, 2013 [135]	***	**	***	8
Gozal, 2014 [136]	****	**	***	9
Kheirandish-Gozal, 2014 [137]	****	**	***	9
Ye, 2015 [138]	***	**	***	8
Huang, 2016 [139]	***	*	***	7
Smith, 2017 [3]	****	**	***	9

Each asterisk denotes 1 point.

**Table 12 life-11-00362-t012:** The results of trim-and-fill method.

Biomarker	Sample	Value	Studies Trimmed	Fixed-Effects			Random-Effects			Q Value
				Point Estimate	Lower Limit	Upper Limit	Point Estimate	Lower Limit	Upper Limit	
hs-CRP	Plasma	Observed	-	0.11402	0.09794	0.13009	0.11531	0.06459	0.16603	70.45500
		Adjusted	0	0.11402	0.09794	0.13009	0.11531	0.06459	0.16603	70.45500
	Serum	Observed	-	0.04957	0.04690	0.05224	0.08898	0.06872	0.10924	887.14388
		Adjusted	14	0.04878	0.04611	0.05144	0.05914	0.03838	0.07990	1122.18105
CRP	Plasma	Observed	-	0.13087	0.09629	0.16545	0.10737	−0.25635	0.47110	675.22529
		Adjusted	1	0.12546	0.09089	0.16004	−0.31732	−0.72147	0.08682	882.03388
	Serum	Observed	-	0.09409	0.07958	0.10861	0.30390	0.21652	0.39129	633.80855
		Adjusted	13	0.05547	0.04154	0.06941	0.08310	−0.01114	0.17735	1132.33477

Abbreviations: CRP, C-reactive protein; hs-CRP, High‑sensitivity CRP.

## Data Availability

Data made available upon request to experts in the field.

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
