# Peer review of "Evaluation of Blood Levels of C-Reactive Protein Marker in Obstructive Sleep Apnea: A Systematic Review, Meta‐Analysis and Meta-Regression"

_life, 2021, doi:10.3390/life11040362_

Round 1
Reviewer 1 Report
Thank you very much for allowing me to review this manuscript. The manuscript provides an updated systematic review and meta-analysis on the association of blood levels of C-reactive protein marker in obstructive sleep apnea. The process and methods of the review is well described allowing readers to replicate it. There are few minor points to address:
- “The review process followed the Preferred Reporting Items for Systematic Reviews and Meta-Analyses (PRISMA) guidelines (23)” . This is imprecise. Prisma guidelines are meant for reporting in systematic reviews and meta-analyses, not for the process of doing the systematic review.
2.The rational behind the criteria of inclusion and exclusion of studies should be included
- “There are no other systematic diseases such as diabetes,” the author mean systemic diseases;
4.“The quality of the studies included in the analysis was evaluated by one author (M.S) using the Newcastle-Ottawa Scale (NOS); nine was a total score of each study (24).” The author mean nine was the maximum total score of each study …
- The authors should include a suggestion of how the study findings contribute to improvement of investigation and management of OSA
Author Response
Thank you very much for all your kind efforts. Please see the detailed point-by-point-response attached as a separate file.
Again, thank you very much for the care devoted to review the manuscript.

Reviewer 2 Report
Meta-analyses and systematic review are well done.
However I have some remarks:
Please explain further, why in adults hs-CRP and CRP were analyzed separately, but combined in children.
In the results section reasons for exclusion are shown in section 3.1 and in figure 2. Please remove redundant information either from the figure or from the text.
A subgroup analysis with regard to cigarette smoking would have been interesting, as smoking can also increase CRP levels. Please include data on smoking status, if possible.
A sensitivity analysis should also be performed on results in children.
Please discuss not only significance, but also effect size in section 4.
Author Response
Thank you very much for all your kind efforts. Please see the detailed point-by-point-response attached as a separate file.
Again, thank you very much for the care devoted to review the manuscript.

This manuscript is a resubmission of an earlier submission. The following is a list of the peer review reports and author responses from that submission.